# Synaptically silent sensory hair cells in zebrafish are recruited after damage

Qiuxiang Zhang[1], Suna Li[1], Hiu-Tung C. Wong[1], Xinyi J. He [1], Alisha Beirl[1], Ronald S. Petralia[2], Ya-Xian Wang[2] & Katie S. Kindt [1]

Analysis of mechanotransduction among ensembles of sensory hair cells in vivo is challenging in many species. To overcome this challenge, we used optical indicators to investigate mechanotransduction among collections of hair cells in intact zebrafish. Our imaging reveals a previously undiscovered disconnect between hair-cell mechanosensation and synaptic transmission. We show that saturating mechanical stimuli able to open mechanically gated channels are unexpectedly insufficient to evoke vesicle fusion in the majority of hair cells. Although synaptically silent, latent hair cells can be rapidly recruited after damage, demonstrating that they are synaptically competent. Therefore synaptically silent hair cells may be an important reserve that acts to maintain sensory function. Our results demonstrate a previously unidentified level of complexity in sculpting sensory transmission from the periphery.

[1] Section on Sensory Cell Development and Function, NIDCD/National Institutes of Health, Bethesda, MD 20892, USA. [2] Advanced Imaging Core, NIDCD/National Institutes of Health, Bethesda, MD 20892, USA. Correspondence and requests for materials should be addressed to K.S.K. (email: katie.kindt@nih.gov)

Within the nervous system, a surplus of circuits, neurons, and synapses provide anatomical redundancy[1–3]. Currently it is unclear whether all neurons and synapses function simultaneously in vivo, and whether redundancy is built in to protect against information loss or neuronal stress. In the inner ear, the activity profile of individual sensory cells is well characterized[4], but within ensembles of primary sensory cells it is not known whether all cells and synapses function together to encode sensory information in vivo.

Hair cells are the sensory cells of the inner ear, and are also present within the lateral-line organs of aquatic vertebrates. Hair cells in the inner ear function to detect sound and vestibular cues, and in the lateral line are used to detect local fluid flow. Hair cells have two distinct functional compartments, located at their apical and basal ends. At the apex, stimuli deflect mechanosensory bundles, open mechanically gated channels, and allow the influx of $K^+$ and $Ca^{2+}$ which depolarizes the hair cell[5]. This depolarization is graded and leads to a voltage change that ultimately activates presynaptic voltage-gated $Ca^{2+}$ channels ($Ca_V1.3$) at the base of the cell, initiating localized $Ca^{2+}$ influx and vesicle fusion at the synapse[6]. While numerous ex vivo studies have demonstrated that this activity profile represents the fundamental framework underlying mechanotransduction in individual hair cells[4], it is not known how a population of hair cells functions in vivo to transmit sensory stimuli.

To understand the functional properties of both individual and populations of hair cells in their native environment, we examined hair cells located in the sensory organs (neuromasts) of the zebrafish lateral-line system[7,8]. Within a neuromast, hair cells can be easily stimulated together and functionally assessed in toto. In addition, using genetically encoded indicators, the activity of all hair cells within a neuromast organ can be imaged simultaneously[9]. The anatomical composition of primary, posterior lateral-line neuromasts is well defined. In each neuromast, there are two populations of hair cells with bundles polarized to respond to stimuli directed in either an anterior or posterior direction[9,10]. At the base of the neuromast, each hair cell has on average three presynapses or 'ribbons' that tether synaptic vesicles at the active zone near $Ca_V1.3$ channels[11]. Postsynaptically, each neuromast organ is innervated by multiple afferent neurons. Each afferent neuron contacts nearly all hair cells of the same polarity, and each hair cell can be contacted by more than one afferent neuron[12]. Overall this anatomy describes a sensory system stacked with anatomical redundancy at many levels—multiple hair cells per polarity, synapses per hair cell, and postsynaptic afferent contacts per hair cell. Therefore, the lateral-line system is poised to address the functional consequence of anatomical redundancy and reveal how a population of hair cells detects and transmits sensory stimuli in its native environment.

For our study, we used optical indicators and cutting-edge imaging methods to simultaneously monitor mechanosensation in all mechanosensory bundles, synaptic transmission at all synapses, or activities at all postsynaptic sites within a neuromast unit. We show that when hair cells are stimulated together, although all hair cells within a neuromast organ are mechanosensitive, the majority of them are synaptically silent, with no presynaptic $Ca^{2+}$ influx, vesicle fusion, or associated postsynaptic activity. Our genetic results indicate that lack of innervation does not alter the proportion of synaptically silent hair cells. Our pharmacological results indicate that networks of glia-like, non-sensory supporting cells may impact presynaptic activity by regulating the intracellular $K^+$ ($[K^+]_{in}$) level in hair cells. We used hair-cell voltage and $Ca^{2+}$ imaging to demonstrate that while high $K^+$ stimulation can depolarize all hair cells within a neuromast, this stimulation is unable to activate $Ca_V1.3$ channels in silent cells. Interestingly, silent cells can be rapidly recruited after damage, perhaps to protect against information loss. Overall, our work demonstrates that sensory hair cells in neuromasts function with many silent connections.

## Results

**Disconnect between mechanosensation and synaptic $Ca^{2+}$.** Similar to neuronal activity, $Ca^{2+}$ changes can be used as a readout of activity in hair cells[9,13,14]. To understand the functional contribution of all the individual hair cells in a neuromast organ, we used a transgenic zebrafish line expressing a hair-cell-specific, plasma membrane-targeted $Ca^{2+}$ indicator GCaMP6s-CAAX[10,15] (Fig. 1b, c, Supplementary Fig. 1a, b). In response to mechanical stimuli, this line can be used to examine $Ca^{2+}$ influx dependent on mechanosensitive ion channels in apical mechanosensory bundles[5,16]. It can also be used to detect subsequent opening of presynaptic, voltage-gated $Ca^{2+}$ channels ($Ca_V1.3$) at the base of hair cells[6,13]. Pharmacological controls confirmed that this indicator line reliably detects hair-cell $Ca^{2+}$ signals within these two distinct functional compartments (see methods, Supplementary Fig. 2).

Using this hair-cell GCaMP6s line, we used a fluid jet to stimulate all hair cells simultaneously (see Methods, Supplementary Fig. 3a–h), and examined $Ca^{2+}$ signals in all mechanosensory bundles or at all presynaptic sites within a neuromast organ by imaging at either the apex or base, respectively (Fig. 1b–g, Supplementary Fig. 1a, b). Upon stimulation, we observed that nearly all hair cells in each neuromast (~90%, $n = 12$ neuromasts) were mechanosensitive and had robust $Ca^{2+}$ influx in their apical hair bundles (example, Fig. 1d, e). In these same cells, we also examined presynaptic $Ca^{2+}$ responses at the base of the cell (presynaptic sites or ribbons marked using a *ribeye b-mCherry* transgenic line; Fig. 1c, Supplementary Fig. 1b). Although we were able to detect robust presynaptic $Ca^{2+}$ signals with focal hotspots located at presynapses[13] (example, Fig. 1f, g), we were only able to detect robust, presynaptic $Ca^{2+}$ signals in a subset of hair cells using the same stimulus. When $Ca^{2+}$ signals were examined over several hours, the same subset of hair cells had presynaptic $Ca^{2+}$ responses over multiple trials spanning several hours (example, Supplementary Fig. 3i–m, $n = 6$ neuromasts).

We examined datasets of stable, presynaptic $Ca^{2+}$ hotspots more closely by placing an ~1.5 μm region of interest (ROI) encompassing the presynapse in the GCaMP6s channel in cells with (example, cell 1 in Fig. 1h) and without (example, cell 2 in Fig. 1h) robust presynaptic $Ca^{2+}$ signals. In cells without robust $Ca^{2+}$ signals, $Ca^{2+}$ signals on and off the presynaptic ribbon were indistinguishable (example, cell 2 in Fig. 1i). In contrast, in cells with robust $Ca^{2+}$ signals, the response at individual presynaptic ribbons was ~80% larger compared to off ribbon sites (example, cell 1 in Fig. 1i; on ribbon: 113.90 ± 21.42%, off ribbon: 23.11 ± 3.93%, $p = 0.0002$, $n = 18$ cells, paired $t$-test). This suggests that mechanosensation does not activate $Ca_V1.3$ channels at presynapses in all cells.

Amongst cells with (active) and without (silent) presynaptic $Ca^{2+}$ signals, we observed a slight, yet significant correlation between the magnitude of the mechanosensory $Ca^{2+}$ response versus the presynaptic $Ca^{2+}$ response (Fig. 1e, g, and j: $R^2 = 0.26$, $p = 0.001$, $n = 94$ cells). We examined the relationship between mechanosensory and presynaptic $Ca^{2+}$ responses more closely by using a presynaptic GCaMP6s $\Delta F/F$ threshold of 20%, the approximate off-ribbon response magnitude, to separate active and silent cells (Fig. 1j, gray bar). Using this metric, we observed that mechanosensory $Ca^{2+}$ responses were on average significantly larger in active cells (Fig. 1k). Although the magnitude of the mechanosensitive response was on average larger in active cells, the presence or magnitude of the mechanosensitive response

did not always dictate whether there was a corresponding presynaptic Ca$^{2+}$ response (see example Fig. 1e versus g and spread in k). Overall these results indicate that although all hair cells are able to detect stimuli, only a subset of hair cells has consequent presynaptic Ca$^{2+}$ signals.

**Vesicle fusion is only detectable in a subset of hair cells.** Previous work has shown that, similar to neurons, fusion of synaptic vesicles in sensory hair cells is tightly linked to presynaptic Ca$^{2+}$ influx[6]. Our observation that not all hair cells in neuromast organs had robust presynaptic Ca$^{2+}$ signals in response to

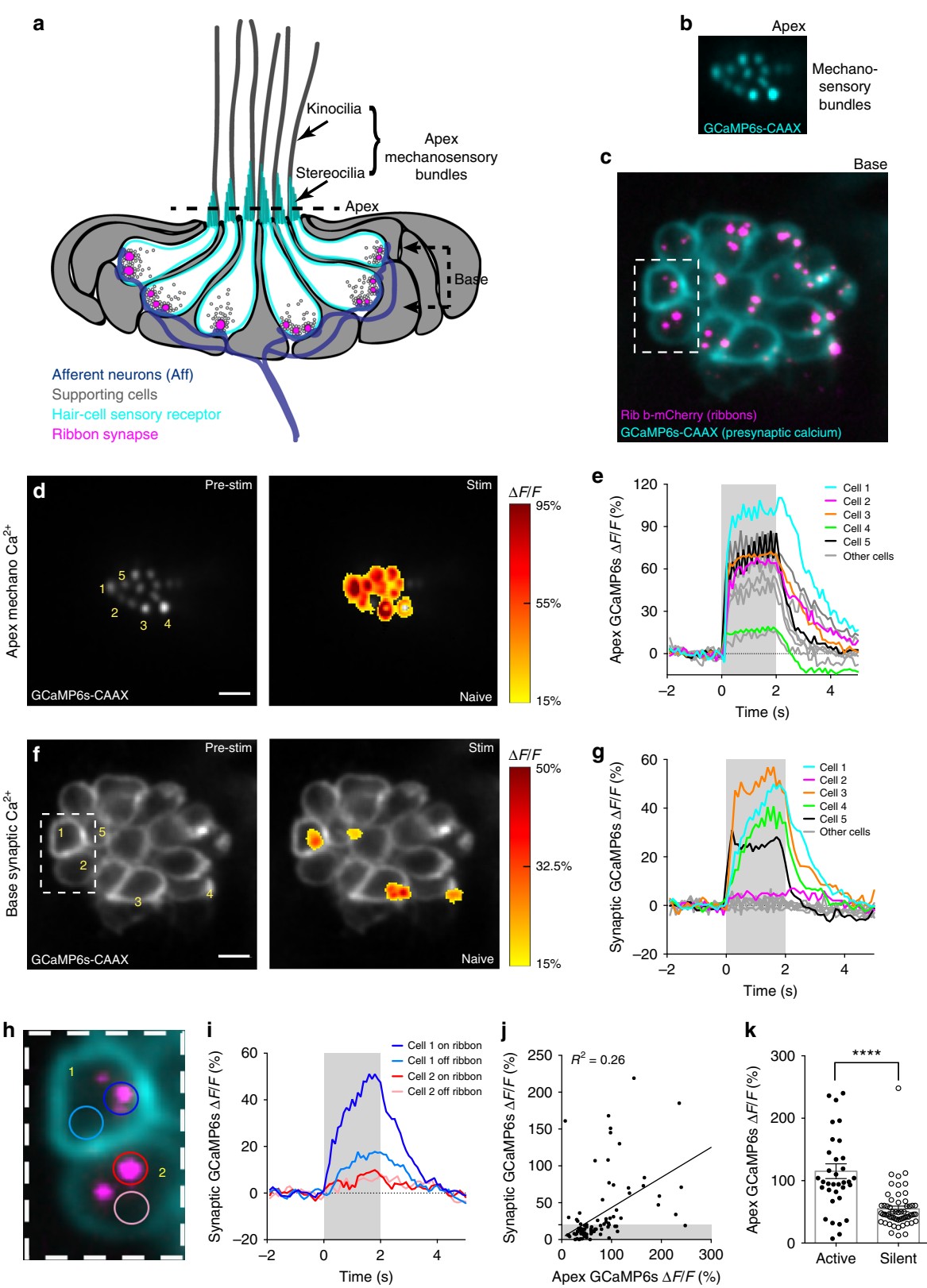

mechanical stimulation suggests that vesicle fusion may not occur at all hair-cell synapses. To assay vesicle fusion in response to stimulation in hair cells, we created a transgenic zebrafish line expressing SypHy, an indicator of vesicle fusion[17] (Fig. 2a, Supplementary Fig. 1e, g).

In response to a stimulus that initiates apical, mechanosensation-dependent $Ca^{2+}$ influx in all hair bundles, we observed that only a subset of hair cells (~30%) had robust increases in SypHy signal at presynapses (example, Fig. 2b–e, white ROIs in d, e). Consistent with our $Ca^{2+}$ imaging results, many cells were presynaptically silent, with no detectable SypHy signal (example, Fig. 2b–e, orange ROIs in d, e). While SypHy signals were detected in significantly fewer hair cells after application of the L-type $Ca^{2+}$ channel antagonist isradipine (Fig. 2f, ~85% reduction), no additional cells displayed SypHy or presynaptic GCaMP6s $Ca^{2+}$ signals after application of the L-type $Ca^{2+}$ channel agonist Bay K (Fig. 2f, Supplementary Fig. 2k–o). Overall, our SypHy imaging reveals that $Ca^{2+}$-dependent vesicle fusion at presynapses occurs only in a subset of hair cells in neuromast organs.

**Stimulus type or age do not alter the number of active cells**. A disparity between mechanosensation and presynaptic vesicle fusion was unexpected because extensive work on hair cells predicts that opening of mechanosensitive ions channels depolarizes the hair cell and triggers $Ca^{2+}$ channel opening and vesicle fusion at the synapse[4–6]. Therefore, we assayed whether inadequate stimulation or cellular immaturity could account for this disparity.

To test whether alternative types of sensory stimuli, or stronger stimuli, could engage silent hair cells, we tested a variety of stimuli including: low (5 Hz for 2 s) and high (50 Hz for 2 s) frequency, short (2 s) and long (6 s) square steps, a ramped step (ramp to max over 2 s) at a saturating stimulus (~5 μm deflection of kinocilial tips), as well as an intense 2 s (square step) or 6 s (5 Hz) stimulus (10 μm deflection of kinocilial tips). While we observed fewer hair cells with vesicle fusion and presynaptic $Ca^{2+}$ responses during a 50-Hz stimulus (Fig. 2g, j), for all other stimuli tested, a similar percentage of hair cells per neuromast had vesicle fusion (Fig. 2g), and the same subset of hair cells had robust presynaptic $Ca^{2+}$ signals (example, Fig. 2i–n).

If alternative stimuli cannot recruit silent cells then it is possible that not all hair cells within the neuromast are mature and have intact synapses. In zebrafish, the lateral line is functional by day 4, before the age at which we performed the majority of our experiments, and each neuromast organ consists of a large collection of mature hair cells as well as immature cells[9]. We used

immunohistochemistry and found that both active and silent hair cells had a similar number of synapses and nearly all synapses stained positive for both presynaptic Ribeye b and postsynaptic MAGUK (Supplementary Fig. 4h–k). This suggests that the majority of mechanosensitive hair cells have intact synapses at the ages we examined.

In addition to this morphological analysis, we also examined vesicle fusion at multiple ages of neuromast-organ development (day 2–6, 13). From day 2 to day 13, the primary posterior lateral-line neuromasts rapidly add hair cells and this growth plateaus around days 5–6 (Fig. 2h, gray bars). Using SypHy, we found that regardless of developmental age (day 2 to 13) or number of hair cells per neuromast (eight cells at day 2, over 20 cells at day 13), each organ had a similar percentage of hair cells with vesicle fusion (Fig. 2h, black bars). Overall our analyses indicate that independent of stimulus type or developmental age, neuromast organs exist in a homeostatic state where only a subset of the hair cells support presynaptic activity.

**Presynaptic activity correlates with afferent activity**. Our presynaptic $Ca^{2+}$ and vesicle fusion measurements in hair cells revealed that not all hair cells have active presynapses. To confirm that only active cells are transmitting sensory stimuli, we created a GCaMP6s transgenic line to examine $Ca^{2+}$ responses in postsynaptic afferents (Fig. 3a, d and Supplementary Fig. 1f, h). Using this indicator line, upon hair-cell stimulation, we were able to detect robust postsynaptic $Ca^{2+}$ responses (example, Fig. 3b). Postsynaptic $Ca^{2+}$ signals initiated adjacent to presynaptic ribbons (example, Fig. 3c), and similar to our SypHy data, we observed that only a subset (~35%) of hair cells was associated with postsynaptic activity (Fig. 3b–e, active cells: white ROIs in d).

Extensive research in neurons and hair cells has shown that vesicle fusion and associated postsynaptic activity are intrinsically linked to the amount and duration of presynaptic $Ca^{2+}$ influx[6,18–20]. Therefore, to directly test whether presynaptic $Ca^{2+}$ influx, vesicle fusion, and afferent activity were associated with the same subset of hair cells within a given neuromast organ, we performed two-color functional imaging. For these experiments, we used a transgenic line that expresses the red-shifted cytosolic $Ca^{2+}$ indicator RGECO1 in hair cells,[13] in combination with either SypHy (green) in hair cells, or GCaMP6s (green) in afferent neurons (Fig. 3g and Supplementary Figs. 1g, h, 5a). We found that RGECO1 $Ca^{2+}$ responses in hair cells associated with afferent activity were larger (70%) than the responses in cells without afferent activity (Fig. 3m, example Fig. 3g–l). Similarly, RGECO1 $Ca^{2+}$ responses were significantly larger (77%) in hair cells with vesicle fusion (Supplementary Fig. 5g, example

**Fig. 1** Hair-cell mechanosensation and presynaptic $Ca^{2+}$ activities. **a** Cartoon neuromast organ illustrating hair cells (cyan) with apical mechanosensory bundles, basal presynaptic ribbons (magenta), supporting cells (gray), and the associated afferent processes (blue). Bundles made up of stereocilia (cyan) and kinocilia (gray) are oriented to respond to either anterior or posterior stimuli. **b, c** Images of a double transgenic neuromast expressing GCaMP6s-CAAX (labels apical and basal hair-cell membranes) and Ribeye b-mCherry (labels presynaptic ribbons) in hair cells. A top-down cross-section reveals mechanosensory bundles (**b**), or the synaptic plane (**c**). **d, f** Functional $Ca^{2+}$ imaging acquired from the neuromast in **b, c** during a 2-s 5 Hz (anterior–posterior directed square wave) mechanical stimulus that stimulates all hair cells. Spatial patterns of GCaMP6s $Ca^{2+}$ signals during stimulation (right panels) are colorized according to the $\Delta F/F$ heat maps and superimposed onto a baseline GCaMP6s images (left panels). At the apex, robust mechanosensitive $Ca^{2+}$ signals are detected in nearly all bundles (**d**), while only a subset of hair cells shows robust presynaptic $Ca^{2+}$ signals at their base (**f**). **e, g** Temporal curves of mechanosensitive (1.5 μm ROIs) or presynaptic (3 μm ROIs) $Ca^{2+}$ responses of all 16 hair cells in this example, highlighting four cells with robust presynaptic $Ca^{2+}$ signals (cells 1, 3, 4, 5, labeled in **d, f**). **h** Enlarged image of the dashed box in **c** and **f** highlights an active (cell 1) and a silent (cell 2) cell. **i** Using 1.5 μm ROIs (**h**), in cell 1, the $Ca^{2+}$ signals are larger on ribbons compared to off ribbon (blue traces) while in cell 2, $Ca^{2+}$ signals off and on the ribbon are similar (red traces). **j** In response to an unsaturated stimulus, there is a slight correlation between presynaptic and mechanosensitive $Ca^{2+}$ signals, $R^2 = 0.26$, $n = 94$ cells. **k** Using a threshold of $\Delta F/F = 20\%$, the active and silent cells in **j** were separated. Mechanosensitive responses were on average larger in active cells (115.1% ± 11.79, $n = 35$) versus inactive cells (54.71 ± 4.44, $n = 59$). A Wilcoxon test was used in **k**; ***$p < 0.0001$. Scale bars = 5 μm

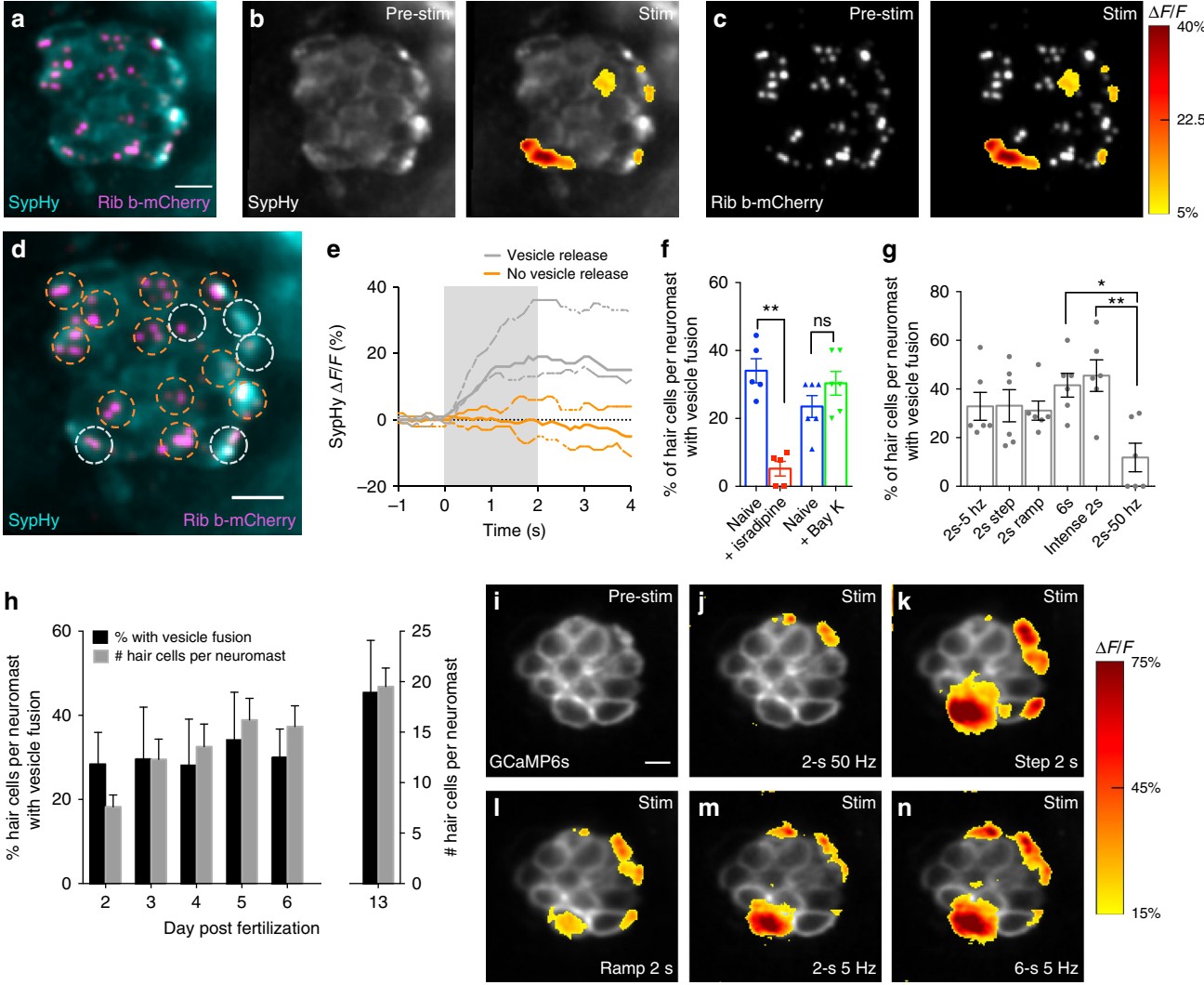

**Fig. 2** Presynaptic vesicle fusion occurs in a subset of hair cells. **a** Neuromast hair cells expressing SypHy (vesicle fusion, cyan) and Ribeye b-mCherry (ribbons, magenta). **b**, **c** Spatial patterns of SypHy signals during a 2-s 5 Hz stimulation (right panels) are colorized according to the $\Delta F/F$ heat map and superimposed onto a baseline SypHy image (**b**, left panel) or onto an image of ribbons (**c**, left panel). Only a subset of the hair cells display vesicle fusion. **d** Dashed circles indicate ROIs (3 μm diameter) used to detect SypHy signals from example in **a–c**. **e** Plot of SypHy signals (mean with upper and lower limits) from cells in **d** from five cells with (white) and eight cells without (orange) vesicle fusion. **f** The % of hair cells per neuromast with vesicle fusion (naïve blue, 34.04% ± 3.55, $n = 5$ neuromasts) decreases after 10 μM isradipine treatment (red, 5.20% ± 2.16, $p = 0.007$), but is not altered (naïve blue, 23.50% ± 3.20, $n = 6$ neuromasts) after 10 μM Bay K treatment (green, 30.33% ± 3.48, $p = 0.10$). **g** The % of hair cells per neuromast with vesicle fusion in response to different stimuli, $n = 6$ neuromasts per stimulus. **h** The % of hair cells per neuromast with vesicle fusion does not vary during development despite the increase in hair-cell number, $n = 6$ neuromasts per age. The % of cells with a SypHy response at day 2 (28.57% ± 3.01) is not different from day 13 (45.47% ± 5.46), $p = 0.11$. **i–n** The same subset of hair cells has robust presynaptic $Ca^{2+}$ signals in response to a variety of stimuli, $n = 5$ neuromasts. Similar to **g**, in **j**, fewer active cells are observed in response to a 2-s 50 Hz stimulus. A paired t-test was used for comparisons in **f**. A one-way ANOVA, df = 30 or 2-way ANOVA, df = 25, with post-hoc Tukey's test to correct for multiple comparisons were used in **g** and **h**, respectively; *$p < 0.05$, **$p < 0.01$. Scale bars = 5 μm

Supplementary Fig. 5a–f). Overall, these results confirm that active hair cells with strong presynaptic $Ca^{2+}$ responses are indeed the same subset of cells with vesicle fusion and associated postsynaptic activity.

**Active cells are not maintained by neuronal innervation**. Our functional imaging demonstrated that within a neuromast organ, only a subset of hair cells is synaptically active. But how is the subset of active cells maintained or setup? One possibility is that activity could be maintained by the efferent neurons that make synaptic contacts directly on or near hair cells[8,21–23] (see methods and Supplementary Fig. 6). To examine whether innervation could setup or maintain active cells within a neuromast organ, we

used two approaches: decapitation and genetic ablation of innervation.

We first used decapitation to eliminate feedback from efferent cell bodies located in the brain onto posterior lateral-line neuromasts located along the body of the fish. Although presynaptic responses were slightly reduced after decapitation, the overall pattern of presynaptically active hair cells within a neuromast organ was not altered (Fig. 4a–d, $n = 6$ neuromasts).

For genetic ablation, we examined animals with *neurog1a* knockdown that lack afferent and efferent innervation of lateral-line neuromasts (see methods, Fig. 4e, f)[24]. Compared to wild-type controls, knockdown of *neurog1a* did not alter the

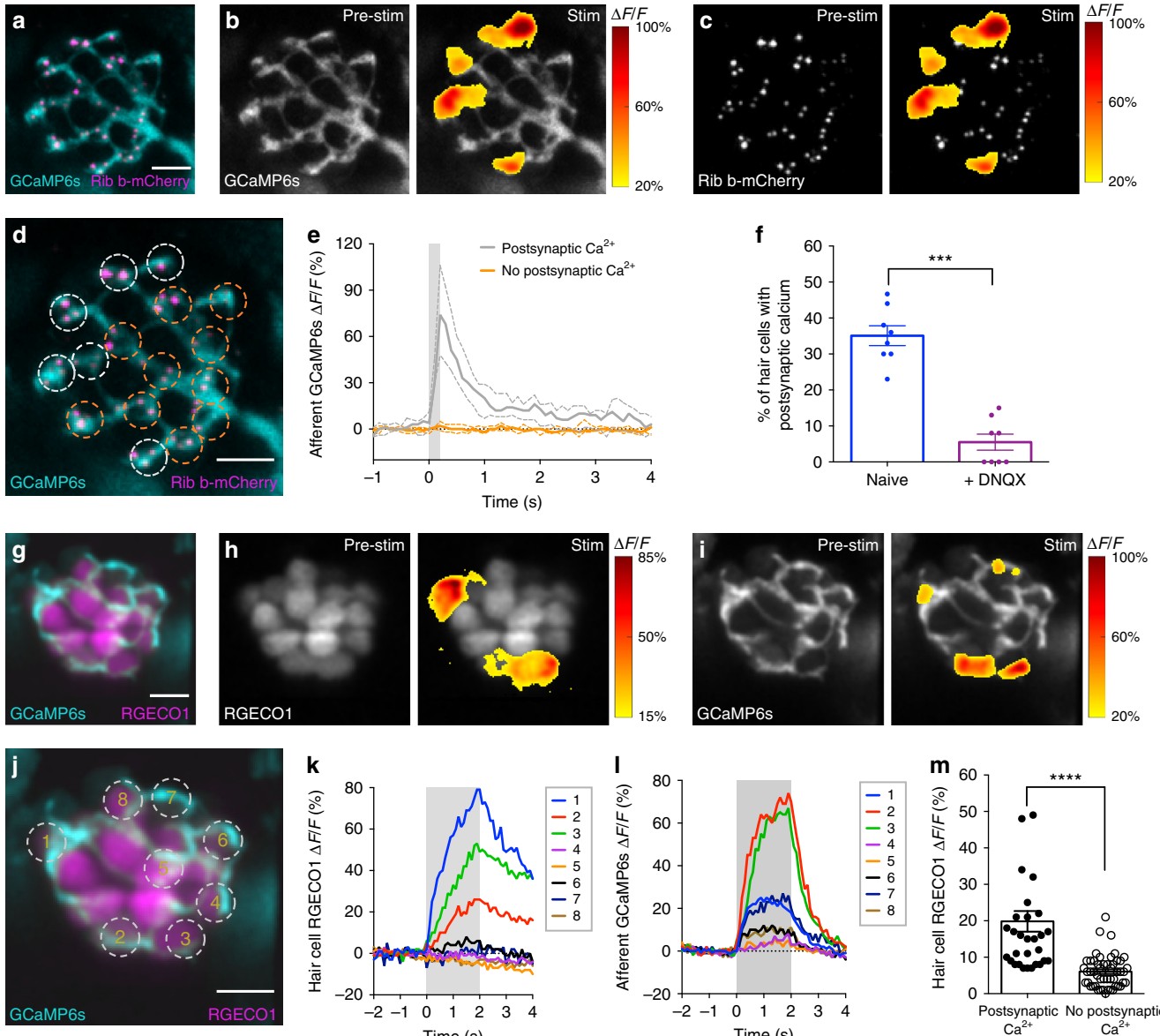

**Fig. 3** Robust hair-cell Ca²⁺ influx corresponds with postsynaptic Ca²⁺ responses. **a** A double transgenic line with afferent neurons expressing GCaMP6s-CAAX to detect postsynaptic Ca²⁺ activity (cyan) and hair cells expressing Ribeye b-mCherry to label ribbons (magenta). **b**, **c** Representative neuromast demonstrating spatial patterns of afferent GCaMP6s signals during a 200-ms step stimulation (right panels). GCaMP6s signals correspond to the $\Delta F/F$ heat map and are superimposed onto the baseline GCaMP6s image (**b**, left panel) or relative to presynaptic ribbons (**c**, left panel). Only a subset of hair cells is associated with postsynaptic Ca²⁺ activity. **d** Dashed circles indicate ROIs (diameter of 3 μm) used to detect afferent Ca²⁺ signals. **e** Plot of the postsynaptic Ca²⁺ signals in the six cells with (white) and 10 cells without (orange) postsynaptic Ca²⁺ activities (mean with upper and lower limits plotted). **f** The percentage of hair cells associated with afferent Ca²⁺ activity (naive blue, 35% ± 2.76) is decreased after 10 μM DNQX treatment (purple, 5.50 ± 2.24) to block postsynaptic AMPA receptors, $n = 8$ neuromasts, $p < 0.0001$. **g** Double transgenic with hair cells expressing RGECO1 (magenta) and afferent neurons expressing GCaMP6s-CAAX (cyan). **h–l** RGECO1 and GCaMP6s responses acquired from the same neuromast organ during a 2-s 5 Hz (anterior–posterior directed square wave) stimulus that activates all hair cells. Hair-cell RGECO1 (**h**, right panel) or afferent GCaMP6s (**i**, right panel) responses during stimulation are superimposed onto the baseline grayscale images (**h**, **i**, left panels). **j–l** 10 ROIs (3 μm) outlined in **j** were used to generate plots of the hair-cell RGECO1 Ca²⁺ (**k**) or GCaMP6s afferent Ca²⁺ (**l**) signals. Hair cells with strong Ca²⁺ influx (cells 1–3, **k**) also have afferent Ca²⁺ signals (**l**). **m** RGECO1 hair-cell Ca²⁺ signals associated with afferent Ca²⁺ signals (19.84% ± 2.38, $n = 31$ cells) were larger than those without (6.04% ± 0.62, $n = 50$ cells), $p > 0.0001$. A paired $t$-test was used in **f**, a Mann–Whitney test was used in **m**; ***$p < 0.001$, ****$p < 0.0001$. Scale bars = 5 μm

magnitude of presynaptic Ca²⁺ responses (Fig. 4g) or the percentage of hair cells with vesicle fusion (Fig. 4h).

Together, our genetic and decapitation results indicate that neither afferent nor efferent innervation is required to setup or maintain active hair cells.

**Supporting cells may regulate K⁺ to enable synaptic activity.** If innervation is not required to set up or maintain synaptic activity, the two remaining cell types within neuromasts that could govern synaptic activity are the sensory hair cells and the glia-like supporting cells (Fig. 1a). In hair-cell organs, hair cells are

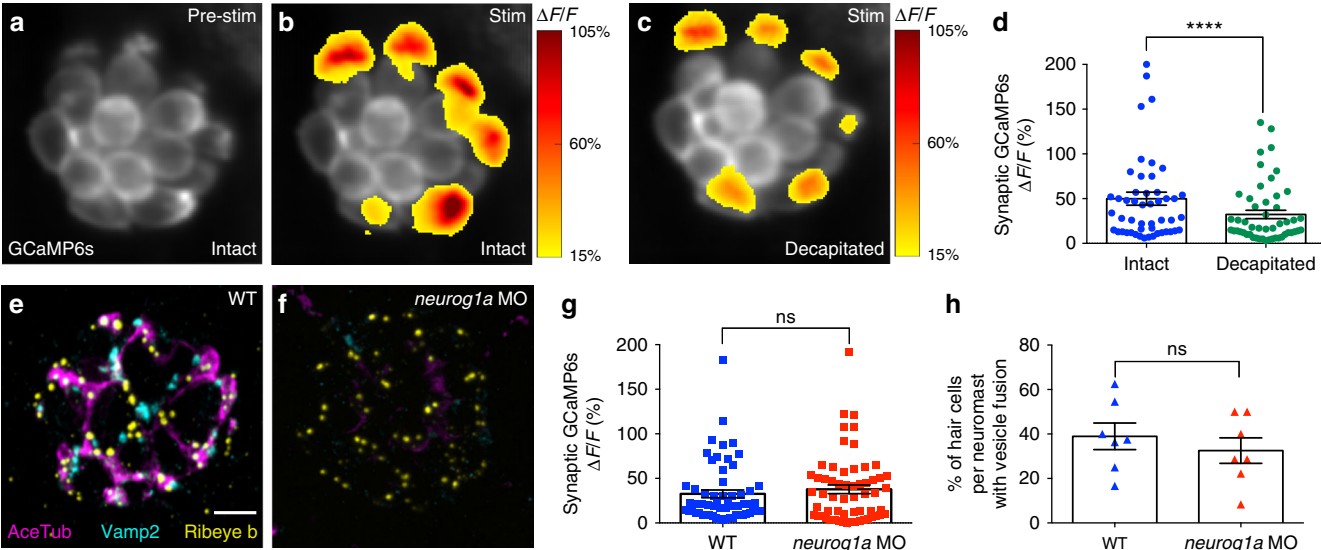

**Fig. 4** The proportion of active cells is not dependent on innervation. **a–c** Presynaptic $Ca^{2+}$ profiles of a representative neuromast in response to a 2-s 5 Hz (anterior–posterior directed square wave) stimulus that activates all hair cells, before (**b**) and after decapitation to eliminate efferent activity (**c**). Spatial patterns of GCaMP6s $Ca^{2+}$ activities during stimulation are colorized according to the $\Delta F/F$ heat map and superimposed onto the baseline GCaMP6s image (**a**). **d** Presynaptic $Ca^{2+}$ responses before (49.90% ± 7.24) and after decapitation (32.31% ± 4.70), $n = 50$ cells, $p < 0.0001$. Immunostaining of a wild-type (WT, **e**) and *neurog1a* morphant (MO, **f**) neuromast. Ribeye b labels presynaptic ribbons (yellow), Acetylated Tubulin (AceTub, magenta) labels afferent neurons, and Vamp2 (cyan) labels efferent neurons. The *neurog1* morphants lack afferent (magenta) and efferent (cyan) innervation. **g** There is no significant difference in the magnitude of the presynaptic $Ca^{2+}$ responses between the WT (32.64% ± 4.36) and *neurog1a* morphants (37.73% ± 4.93), $n = 59$ hair cells, $p = 0.79$. **h** There is no significant difference in the percentage of hair cells with vesicle fusion per neuromast between WT (38.94% ± 5.97) and *neurog1a* morphants (32.53% ± 5.75), $n = 7$ neuromasts, $p = 0.45$. A Wilcoxon test was used in **d**, a Mann–Whitney test in **g**, and an unpaired *t*-test in **h**; ****$p < 0.0001$. Scale bars = 5 μm

surrounded from apex to base by supporting cells that form a syncytia that is electrically coupled by gap junctions[25–27]. In the nervous system, gap junction-coupled syncytia of glia function to remove extracellular $K^+$ ([$K^+$]$_{ex}$) and spatially buffer $K^+$ among glia to control neuronal excitability[28–30]. Therefore, we hypothesized that networks of supporting cells could remove [$K^+$]$_{ex}$ and spatially buffer $K^+$ to facilitate presynaptic activity.

We used APG-2, a vital dye-based $K^+$ indicator[31] to examine whether there were differences in [$K^+$]$_{in}$ levels between active and silent cells. We first used the red-shifted $Ca^{2+}$ indicator jRCaMP1a in hair cells to distinguish between active and silent cells (example Fig. 5a) and then we applied APG-2 to measure the relative resting [$K^+$]$_{in}$ among hair cells within each neuromast (example Fig. 5b). By measuring AGP-2 intensity, we found that [$K^+$]$_{in}$ levels were significantly lower (~35%) in active cells compared to silent cells (Fig. 5c). We also used GCaMP6s to measure baseline $Ca^{2+}$ levels in active and silent cells. In contrast to our APG-2 measurements, using GCaMP6s, we were not able to detect a difference in the baseline $Ca^{2+}$ levels at the presynapse between active cells and silent cells (Supplementary Fig. 7m), although the baseline $Ca^{2+}$ levels were slightly elevated in the hair bundles of active cells (Supplementary Fig. 7l). Based on these baseline measurements, relatively low [$K^+$]$_{in}$ is associated with active cells.

Optimal $K^+$ buffering between glia in the CNS requires effective gap junction coupling[32]. In addition, in mammalian supporting cells, the $Cl^-$ channel TMEM16A/Ano1 has been shown to be important for $K^+$ release and hair-cell depolarization during development[33]. Using transmission electron microscopy (TEM), we found that, as in mammals, gap junctions are present between supporting cells within neuromast organs, indicating that this feature is conserved in the zebrafish lateral line (Supplementary Fig. 7a–g). To test whether gap junctions present between supporting cells could regulate hair-cell [$K^+$]$_{in}$ levels, we

measured [$K^+$]$_{in}$ levels before and after application of the gap junction antagonist flufenamic acid (FFA). We observed that after FFA application, [$K^+$]$_{in}$ levels were significantly elevated in hair cells (Fig. 5d, e) as well as in supporting cells (Fig. 5f).

To test whether gap junction coupling could facilitate presynaptic activity by maintaining low [$K^+$]$_{in}$ levels in active hair cells, we applied two distinct classes of gap junction blockers, FFA (Fig. 5g–n, and Supplementary Fig. 7k) and heptanol (Supplementary Fig. 7h–j). We observed that while FFA and heptanol had no impact on mechanosensation (Fig. 5g–i, j and Supplementary Fig. 7h), presynaptic $Ca^{2+}$ signals were reduced by 50% (Fig. 5k–m, n and Supplementary Fig. 7i). In addition, both gap junction channel blockers dramatically reduced the percentage of hair cells with vesicle fusion (Supplementary Fig. 7j, k). We also tested whether block of $Cl^-$ channels could impact synaptic activity. After application of the $Cl^-$ channel blocker T16A(inh)-A01, we observed a 30% reduction in presynaptic $Ca^{2+}$ signals (Supplementary Fig. 7n). Together our $Ca^{2+}$ and $K^+$ imaging results indicate that supporting cells may require both gap junctions as well as $Cl^-$ channels to facilitate presynaptic activity in active cells. Gap junctions may act broadly to buffer $K^+$ and maintain [$K^+$]$_{in}$ levels in hair cells. Interestingly, $K^+$ buffering is only able to maintain low [$K^+$]$_{in}$ levels and facilitate synaptic activity in a subset of cells (Fig. 5b). In this manner, it is possible that non-sensory supporting cells may play a role in encoding sensory stimuli.

**Depolarization does not activate silent synapses.** Although the exact mechanism required to setup or maintain a population of synaptically active hair cells is unclear, our immunohistochemistry results indicate that both active and silent hair cells have pre- and postsynaptic specializations (Supplementary Fig. 4i–k). This suggests that presynaptic $Ca^{2+}$ channels may also be present in all

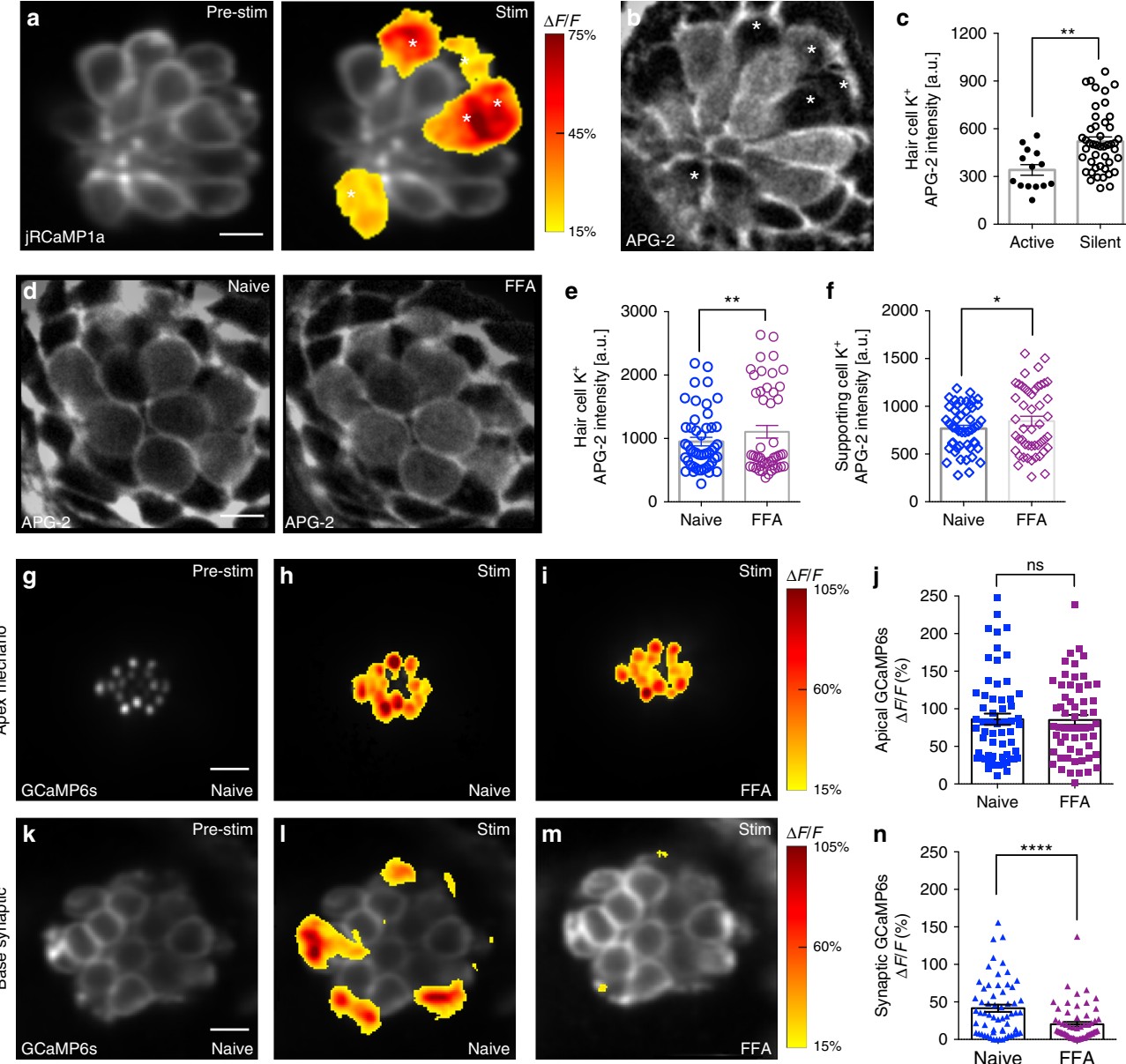

**Fig. 5** Low [K⁺] and gap junctions may facilitate presynaptic activity. **a** Presynaptic Ca²⁺ jRCaMP1a Ca²⁺ signals during a 2-s 5 Hz stimulus. Spatial patterns of jRCaMp1a Ca²⁺ signals during stimulation are colorized according to the $\Delta F/F$ heat maps and superimposed onto a pre-stimulus baseline jRCaMP1a image (**a**, left panel). **b** Image of the same neuromast as **a** after labeling with the K⁺ indicator, APG-2. The active cells in **a** and **b** are marked with asterisks. **c** Quantification of APG-2 intensity shows that active cells (341.1 a.u. ± 33.17, $n = 14$ cells) have lower resting [K⁺]$_{in}$ levels than silent cells (520.1 a.u. ± 28.37, $n = 46$ cells) $p = 0.001$. **d** APG-2 dye labeling before (**d**, left panel) and after (**d**, right panel) FFA treatment to block gap junctions. **e**, **f** Quantification of APG-2 intensity shows 25 µm FFA significantly increases [K⁺]$_{in}$ levels in hair cells (naïve: 950.2 a.u. ± 67.9; after FFA: 1105 a.u. ± 99.7, $n$ = 48 cells, $p = 0.01$) and in supporting cells (naïve: 767.1 a.u. ± 34.4; after FFA: 845.1 a.u. ± 50.3, $n = 48$ cells, $p = 0.015$). **g–i**, **k–m** Mechanosensitive and presynaptic GCaMP6s Ca²⁺ signals within the same cells before and after application of 25 µM FFA. Mechanosensative (**h**) and presynaptic (**l**) Ca²⁺ signals during a 2-s 5 Hz stimulus prior to drug treatment; 25 µM FFA does not alter mechanotransduction (**i**) but decreases presynaptic Ca²⁺ responses (**m**). Spatial patterns of GCaMP6s Ca²⁺ signals during stimulation are colorized according to the $\Delta F/F$ heat maps and superimposed onto baseline GCaMP6s images (**g**, **k**). **j** Quantifications of the mechanosensitive Ca²⁺ signals show no significant differences before (86.10% ± 7.55) and after 25 µM FFA (85.18% ± 6.64), $n = 60$ hair cells, $p = 0.76$. **n** In the same cells as **j**, presynaptic Ca²⁺ signals (41.55% ± 4.85) are significantly decreased after FFA application (20.27% ± 3.07), $n = 60$ cells, $p < 0.0001$. A Mann–Whitney test was used in **c**, a Wilcoxon test was used in **e**, **f**, **j**, and **n**; *$p < 0.05$, **$p < 0.01$, ****$p < 0.0001$. Scale bars = 5 µm

neuromast hair cells. To confirm this, we used immunohistochemistry and observed that ~93% (Fig. 6d) of presynapses within a neuromast are associated with clusters of the voltage-gated Ca²⁺ channel Ca$_V$1.3 (example, Fig. 6a–c). This labeling suggests that although Ca$_V$1.3 channels are present at presynapses in silent cells, mechanosensation is unable to activate these

channels. Why are these channels not activated? One possibility is that in silent cells, mechanosensation does not effectively depolarize hair cells and is unable to activate voltage-gated Ca$_V$1.3 channels. Alternatively, it is possible that in silent cells, despite a significant change in voltage upon mechanosensation, Ca$_V$1.3 channels are unable to be activated. To directly test whether hair-

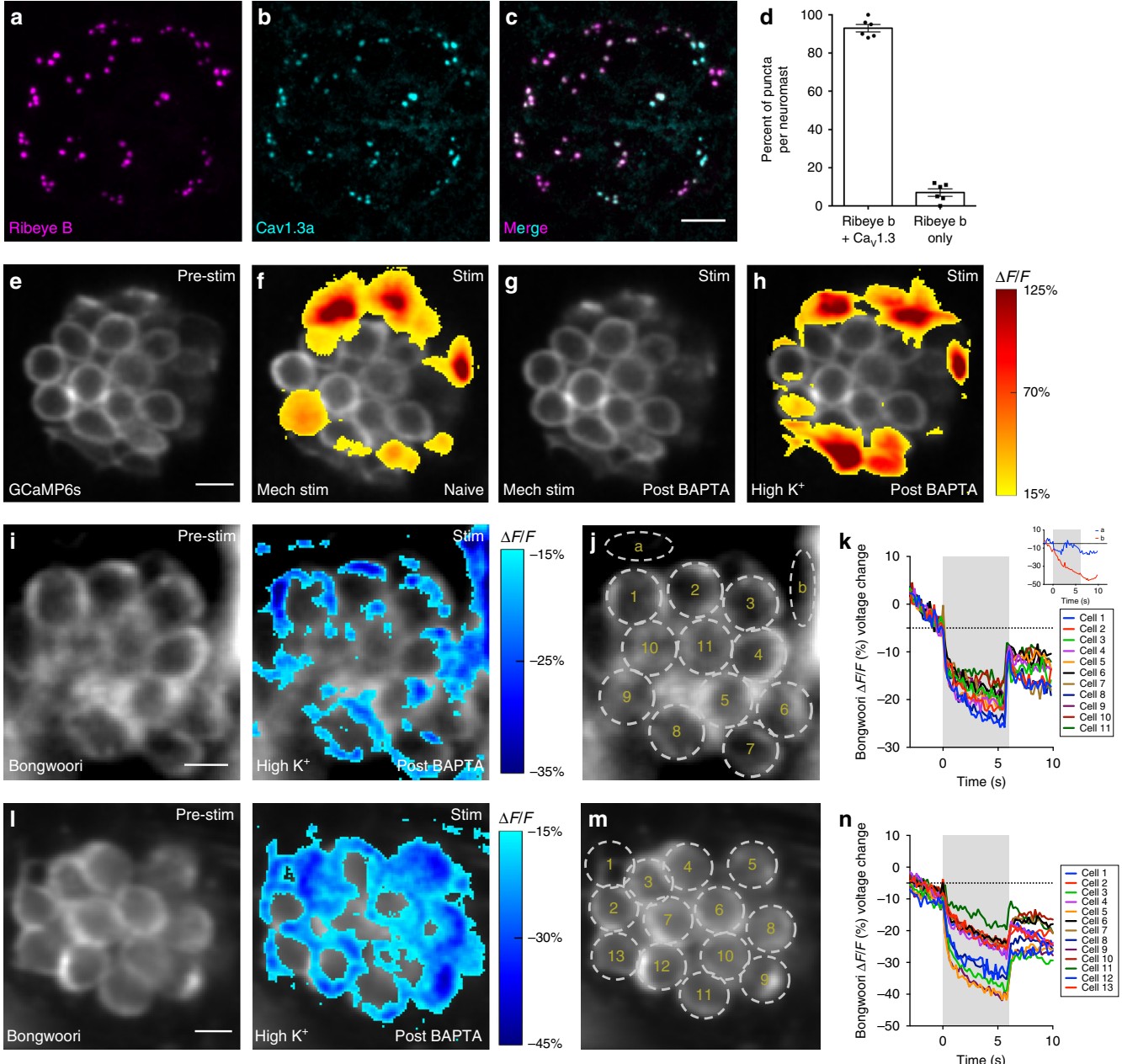

**Fig. 6** Depolarization does not activate $Ca_V1.3$ channels in all cells. **a–c** Example neuromast with immunostain labeling presynaptic ribbons (Ribeye b, magenta) in **a**, and presynaptic $Ca_V1.3a$ channels (cyan) in **b**. Images in **a** and **b** are merged in **c**. **d** Quantification of the percent of Ribeye b positive ribbons per neuromast that colocalize with $Ca_V1.3a$ (93.00% ± 1.93, $n = 10$ neuromasts). **e–h** Representative neuromast demonstrating spatial patterns of hair-cell presynaptic GCaMP6s $Ca^{2+}$ signals during mechanical and high $K^+$ stimulation. Signals are colorized according to the $\Delta F/F$ heat map and superimposed onto baseline GCaMP6s image (**e**). **f** A subset of cells shows presynaptic $Ca^{2+}$ signals in response to a 2-s 5 Hz mechanical stimulus. **g** All mechanical responses are eliminated after BAPTA treatment. **h** After BAPTA treatment, to depolarize hair cells, high $K^+$ was applied (via the fluid jet). High $K^+$ application activates the same cells as the original mechanical stimulus (**f**). **i**, **l** Example neuromasts demonstrating spatial patterns of hair-cell Bongwoori voltage signals during high $K^+$ stimulation. Spatial patterns of Bongwoori voltage signals during high $K^+$ stimulation are colorized according to the $\Delta F/F$ heat map and superimposed onto baseline Bongwoori image (**i**, **l**, left panels). **k**, **n** Using 5 µm ROIs shown in **j**, **m**, all cells show depolarization in response to high $K^+$ application. Inset in **k** demonstrates that changes in Bongwoori signals in the background skin pigment (ROIs **a**, **b**) do no correlate with the stimulus. Scale bars = 5 µm

cell depolarization could activate $Ca_V1.3$ channels in all cells, we bypassed mechanosensation and used high $K^+$ to depolarize hair cells.

For these experiments, we first used a mechanical stimulus and GCaMP6s to identify active hair cells (Fig. 6e, f). Then we applied BAPTA to eliminate all mechanically evoked responses (Fig. 6g, and Supplementary Fig. 2a–e). Finally, we applied high $K^+$ via the

fluid jet to depolarize all hair cells within a neuromast. Upon high $K^+$ application, we observed the same overall pattern of presynaptic $Ca^{2+}$ signals that we observed using a mechanical stimulus (example Fig. 6h, $n = 6$ neuromasts). This indicates either high $K^+$-induced depolarization is unable to activate $Ca_V1.3$ channels in silent cells, or high $K^+$ is unable to depolarize all hair cells. To distinguish between these scenarios, we created a

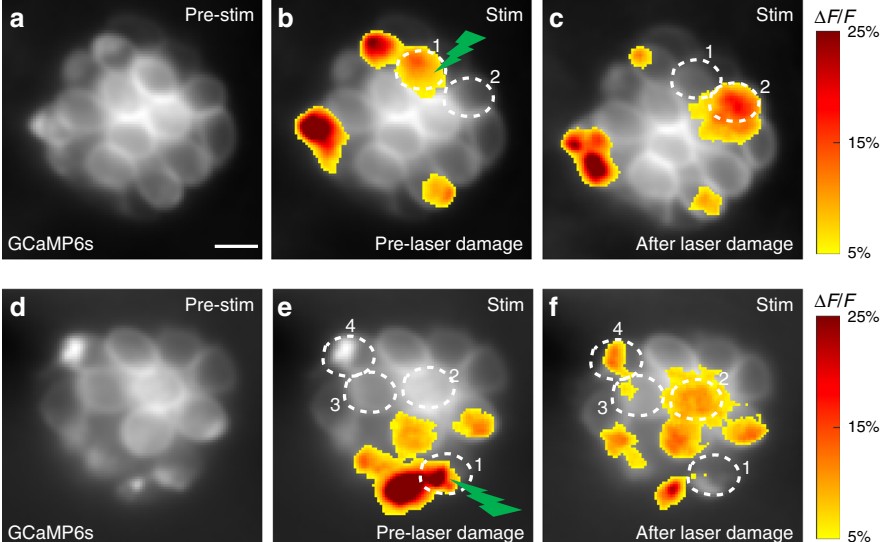

**Fig. 7** Laser damage to an active cell activates silent cells. **a–f** Presynaptic GCaMP6s signals from two representative neuromast organs before and after laser damage. **b**, **e** Presynaptic Ca$^{2+}$ signals during a 2-s step stimulus (anterior and posterior responses are merged in the profile) prior to laser damage. Spatial patterns of GCaMP6s Ca$^{2+}$ signals during stimulation are colorized according to the $\Delta F/F$ heat map and superimposed onto baseline GCaMP6s images (**a**, **d**). **c**, **f** Presynaptic Ca$^{2+}$ signals after high power laser-induced damage to one active cell in each neuromast (green lightning bolt, cell 1 in **b** and **e**). After laser damage, the damaged cell (cell 1) no longer displays a detectable presynaptic Ca$^{2+}$ signal, but new cells now display robust presynaptic Ca$^{2+}$ signals (cell 2 in **c** and cells 2–4 in **f**). Scale bars = 5 μm

transgenic line expressing the genetically encoded voltage indicator Bongwoori in hair cells (Supplementary Fig. 1d) to measure the membrane voltage changes rather than presynaptic Ca$^{2+}$ signals during high K$^+$ application. Using Bongwoori, we found that high K$^+$ application robustly depolarized all hair cells within a neuromast (examples Fig. 6i–n, $n = 7$ neuromasts). Overall using high K$^+$ application, along with hair-cell Ca$^{2+}$ and voltage imaging, our data suggest that hair-cell depolarization may not activate Ca$_V$1.3 channels in silent cells.

**Silent cells can become active after damage to active cells**. Our Ca$^{2+}$ and voltage imaging revealed that Ca$_V$1.3 channels are not engaged in silent cells despite mechanosensation and hair-cell depolarization. From these results, an important question remained—are silent hair cells capable of synaptic activity? To probe this question, we asked whether silent cells could become synaptically competent after active cells were incapacitated.

For these experiments, we first measured presynaptic Ca$^{2+}$ signals to determine which cells were active (Fig. 7a, b, d, e). Then we used a 405-nm laser to damage an active cell (examples, cell 1 in Fig. 7b, e); 10–30 min after laser damage, we re-examined the pattern of presynaptic Ca$^{2+}$ signals in the same neuromast. Surprisingly, in approximately half of the trials ($n = 14/26$ neuromasts), we found that previously silent hair cells were now active (examples, cell 2 in Fig. 7c, and cells 2–4 in Fig. 7f). Looking closely at the change in activity patterns after laser damage ($n = 14$), we observed that in some cases, the new active cells were adjacent to the cell that was damaged (example, cell 2 in Fig. 7a–c), but in other instances, the new active cells were relatively far from the damaged cell (example, cells 2–4 in Fig. 7d–f). In the majority of trials after damage to a single active cell, we observed one new active cell, but in some trials, two or more new cells became active (29%). Of the two functional populations of hair cells within neuromast organs (polarized to respond to either an anterior or posterior directed stimulus), after damage, the new active cell could have the same or opposite polarity as the damaged cell (same direction: 57%, opposite direction: 43%). Previous work has shown that each population of

neuromast hair cells is innervated by a distinct set of afferents[12]. Because damage does not always replace an active cell with a cell of the same polarity, this indicates that relevant wiring or function may not be integrated into the unsilencing.

To further test whether wiring of the neuromast could impact unsilencing, we performed laser damage experiments in *neurog1a* mutants that lack efferent and afferent innervation. Similar to wild-type animals, we observed that silent cells could become active in *neurog1a* mutants following damage to an active cell ($n = 5/7$ trials). This suggests that innervation of hair cells is not required to make this conversion, and may not influence which hair cell changes from a silent to an active state after laser damage. Overall our laser damage experiments indicate that many silent cells are poised and ready for presynaptic activity, and within each organ there is a coordinated effort to maintain a subset of active cells.

## Discussion

A long-standing question in neuroscience is whether each structurally defined synapse is functional. In our study, we have identified a sensory system where under normal operating conditions, the majority of synapses are presynaptically silent despite robust sensory input and depolarization. Overall, our work demonstrates that there is functional heterogeneity among hair cells within the same organ and highlights the complexity underlying hair-cell function in vivo.

Our in vivo approach has shown that, in their native environment, despite all being mechanosensitive, not all neuromast hair cells are synaptically active. This is in contrast to previous ex vivo studies using electrophysiological recordings performed in a number of species—including zebrafish—that demonstrate voltage steps applied to individual hair cells reliably result in synaptic activity[6,34,35]. But how do electrophysiological preparations reliably produce synaptic responses in all cells, while our work identified so many silent cells? One explanation is that in hair-cell preparations for electrophysiology, it is challenging to maintain and stimulate hair cells in their native environment. For example, in the zebrafish lateral line, after electrically isolating a

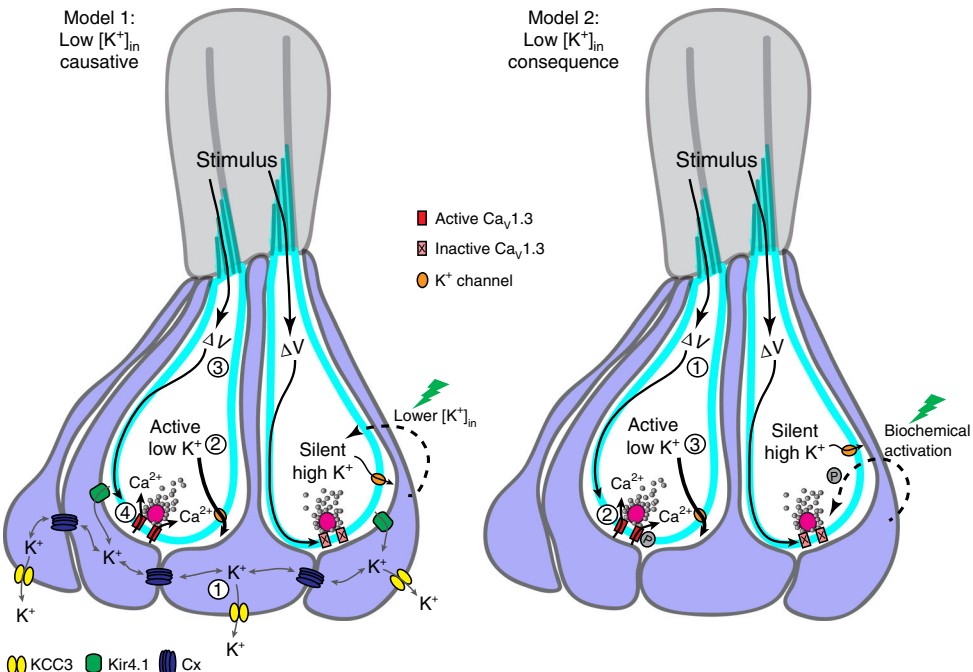

**Fig. 8** Model of presynaptic silencing in neuromast organs. Left-side, Model 1: low $[K^+]_{in}$ is causative for $Ca_V1.3$ channel (red) activation. In active cells, (1) similar to glia, supporting cells may use Kir4.1 channels (green) to take up $[K^+]_{ex}$, gap junctions (Cx, blue) to spatially buffer $K^+$ among syncytia of supporting cells, and a $K^+$-$Cl^-$ cotransporter such as KCC3 (yellow) to clear $K^+$ from supporting cells. This results in (2) lower $[K^+]_{in}$ in active cells. Cells with lower $[K^+]_{in}$ may be at sufficiently depolarized resting membrane potentials where (3) mechanosensation and depolarization ($\Delta V$) is able to (4) activate $Ca_V1.3$ channels. In this model, despite $K^+$ buffering by supporting cells, not all hair cells are able to be maintained with $[K^+]_{in}$ levels low enough to facilitate presynaptic function. Blocking gap junctions elevates hair-cell $[K^+]_{in}$ and presynaptic function is lost in all cells. After laser damage (green lightning bolt), a signaling cascade could lower $[K^+]_{in}$ levels in silent cells, and raise resting membrane potentials into a range suitable for $Ca_V1.3$ channel activation. Right-side, Model 2: low $[K^+]_{in}$ is a consequence of presynaptic activity. This schematic demonstrates that (1) mechanosensation and depolarization ($\Delta V$) lead to (2) $Ca_V1.3$ channel activation in active cells. Presynaptic activity results in (3) lower $[K^+]_{in}$. In this model, $Ca_V1.3$ channels may be inactive due to a lack of biochemical modification, such as phosphorylation (P), as shown in this example. Alternatively, $Ca_V1.3$ channels could also be rendered inactive due to a missing interaction partner or improper assembly at the plasma membrane (these examples not shown). In this example, after laser damage (green lightning bolt), a signaling cascade could promote phosphorylation and lead to $Ca_V1.3$ channel activation

hair cell, $[K^+]_{in}$ and resting membrane potential are maintained at standard concentrations and potentials, respectively[34,35]. If low $[K^+]_{in}$ is important for presynaptic function (Fig. 5c), then optimizing the $K^+$ in the solutions could facilitate the detection of $Ca_V1.3$ currents in all cells recorded. Alternatively, it is possible that it is simply challenging to distinguish synaptically silent cells from an unsuccessful recording, making it difficult to identify this silent population. In the future, it will be critical to examine hair cells in their native environment in other hair-cell organs and species to determine if our results hold true across systems.

In our study, we found that all cells appeared to have properly localized $Ca_V1.3$ channels at presynapses, but only hair cells with low $[K^+]_{in}$ were associated with presynaptic $Ca_V1.3$ channel activity. Studies on these channels indicate that $Ca_V1.3$ channel open probability and open time occur optimally within a membrane voltage range from $-60$ to $-40$ mV[36,37]. Based on this work, and the Nernst equilibrium potential, increased $[K^+]_{in}$ in silent cells may reflect hyperpolarized membrane potentials that place the cell out of this optimal activation range (Fig. 8, model 1). If low hair-cell $[K^+]_{in}$ is an important requirement for presynaptic activity, it remains unclear how $[K^+]_{in}$ heterogeneity is set up within a neuromast organ. From our work, it appears that the surrounding supporting cells may function broadly to maintain low hair-cell $[K^+]_{in}$ levels, but are unable to maintain all hair cells within each neuromast with sufficiently low $[K^+]_{in}$ to facilitate presynaptic activity. $K^+$ buffering is important in hair-cell organs, as increases in $[K^+]_{ex}$ have been observed during stimulation of hair cells[38,39]. As previously proposed, our data

support the idea that gap junction-coupled supporting cells could act to sink $K^+$ away from the basolateral domain of hair cells to preserve their indefatigable presynaptic function[40,41]. Similar to glia, it is possible that supporting cells may use $KIR_{4.1}$ channels, along with the $K^+$-$Cl^-$ co-transporters KCC3 and KCC4, to move $K^+$ in and out of the syncytia[42] (Fig. 8, model 1). In support of this, lesions in *Connexin26* (Cx26 subunits form a gap junction channel), $KIR_{4.1}$ and KCC3/4 genes have all been associated with hearing loss in humans or mice[43–45]. Thus far, our present work investigating the role of supporting cells and gap junctions relied solely on pharmacology, and the compounds that block these channels have other targets[46]. Therefore, while the homologs of $KIR_{4.1}$ and Cx26 are expressed in zebrafish hair-cell organs[47,48], additional genetic analyses are needed to determine if they function in zebrafish supporting cells to regulate $[K^+]_{in}$ and hair-cell excitability in neuromasts.

Alternatively, it is possible that low hair-cell $[K^+]_{in}$ levels in active cells is a byproduct of presynaptic function rather than a prerequisite for function (Fig. 8, model 2). This idea is supported by our hair-cell voltage and $Ca^{2+}$ imaging experiments that demonstrate that high $K^+$ stimulation can depolarize all cells (Fig. 6i–n) but is unable to activate $Ca_V1.3$ channels in silent cells (Fig. 6h). Because it is difficult to correlate voltage indicator intensity with actual resting potentials and voltage changes, it is not clear from these experiments whether an optimal membrane potential was reached for $Ca_V1.3$ channel activation in silent cells. If a sufficient depolarization was indeed achieved, then instead it is possible that no amount of depolarization can activate $Ca_V1.3$

channels in silent cells. If so, then differences in $[K^+]_{in}$ levels or resting membrane potential are not sufficient to explain inactive $Ca_V1.3$ channels. Alternatively, $Ca_V1.3$ channels in silent cells may (1) have different interaction partners, (2) different post-translational modifications, or (3) be improperly localized at the plasma membrane (Fig. 8. model 2). Existing literature indicates that all of these scenarios can impact $Ca_V1.3$ channel activity[49–54]. For example, the $Ca_V1.3$ pore forming α-subunit has numerous interaction partners including: $Ca^{2+}$ binding proteins (CaBPs), Calmodulin (CaM), and various auxiliary sub-units[49,50,53]. All of these interactors can impact $Ca_V1.3$ channel properties. In addition, phosphorylation of $Ca_V1.3$ channels by protein kinase A (PKA) or protein kinase G (PKG) has been shown to promote channel activation[51]. Alternatively, $Ca_V1.3$ channels may not be properly trafficked to the plasma membrane. Studies have shown that molecules such as Harmonin and RIM can impact the trafficking of $Ca_V1.3$ channels to the plasma membrane[52,54]. Additional work is needed to determine if any of these processes could explain why $Ca_V1.3$ channels are not activated in silent cells. Regardless of the mechanism that inactivates $Ca_V1.3$ channels, our laser damage experiments demonstrated that presynaptic unsilencing can occur rapidly (Fig. 7, 10–30 min).

How could damage rapidly alter a silent cell so that it is permissive to $Ca_V1.3$ channel activation? Previous work has demonstrated that there is ATP release after hair-cell damage in the mammalian auditory system[55,56]. Based on this work, one possibility is that in neuromast organs, damage leads to ATP release from hair cells or surrounding supporting cells. Ultimately, an ATP-mediated purinergic receptor signaling cascade could function to promote $K^+$ efflux, lower $[K^+]_{in}$ levels, and unsilence cells (Fig. 8, model 1). Alternatively, ATP-mediated signaling could activate kinases such as PKA to directly phosphorylate $Ca_V1.3$ channels and promote channel activation[57] (Fig. 8, model 2). Future work is needed to address whether ATP or an alternative signaling pathway is needed to unsilence hair cells.

Regardless of the exact mechanism used to unsilence cells or maintain a subset of active cells, we have identified a sensory organ that functions with many silent connections. Based on previous anatomical work, it is possible that silent cells may be yet another mechanism in place to protect against information loss. Previously, three levels of anatomical redundancy have been described within neuromasts. First, each of the two hair-cell populations (responding to either an anterior or posterior directed stimulus[10]) is represented by multiple hair cells. Second, each hair cell has approximately three synapses per hair cell[11] and third, each hair cell is innervated by more than one afferent neuron[12]. Surprisingly, unlike these anatomical redundancies, presynaptic unsilencing after laser damage does not always replace an equivalent hair cell. The unsilenced cell could arise from either of the two hair-cell populations. This suggests that damage acts as a broad cue to unsilence hair cells, without a directive to maintain absolute balance and replacement based strictly on anatomy. Perhaps this is why damage can often (29% of trials) unsilence multiple cells—to ensure cells of the correct polarity are replaced.

The presence of silent cells suggests that not all cells are needed for proper lateral-line function and that silent cells exist as backups. In support of this, previous work found that after moderate damage of hair cells in the lateral line, no quantifiable behavioral deficit was observed[58]. Therefore, anatomical redundancy and presynaptic silencing may work together to create a sensory system that functions with the minimum number of cells and connections. This system could be beneficial in several ways. First, if all cells are not needed, silencing the majority of cells

could prevent unnecessary energy loss and cellular stress. Second, silent cells could rapidly backup cells lost after damage–this is especially important in sensory systems which are critical for larval zebrafish survival[59]. Although it is not yet known, it will be exciting to see if our data are generalizable to other hair-cell organs and model systems.

## Methods

**Zebrafish husbandry and strains.** Zebrafish (*Danio rerio*) were grown at 30 °C using standard methods. Larvae were raised in E3 embryo medium (5 mM NaCl, 0.17 mM KCl, 0.33 mM $CaCl_2$, and 0.33 mM $MgSO_4$, pH 7.2). Zebrafish work performed at the National Institute of Health (NIH) was approved by the Animal Use Committee at the NIH under animal study protocol #1362-13. Larvae were examined at 2–6 days post fertilization (dpf) unless stated otherwise. At these ages, sex cannot be predicted or determined, and therefore sex of the animal was not considered in our studies. Previously described mutant and transgenic zebrafish strains used in this study include: Tg(-6myo6b:GCaMP6s-CAAX)$^{idc1Tg}$, Tg(-6myo6b:Ribeye b-mCherry)$^{idc3Tg}$, Tg(-6myo6b:RGECO)$^{vo10Tg}$, and TgBAC(neurod: GFP)$^{nl1}$ [13,60].

To eliminate efferent and afferent neurons that innervate posterior lateral-line neuromasts, we used a morpholino antisense oligonucleotide (MO, Gene Tools) against *neurog1a*[24]. A start codon MO for *neurog1a* was used: MP-5′ ATCGGAGTATACGATCTCCATTGTT3′. The MO was diluted in water and 1 nL of a 900 µM solution was injected into 1-cell stage embryos. At this concentration, ~50% of the *neurog1a* morphant larvae had no acoustic startle response at day 5, consistent with a lack of afferent innervation. After functional imaging, to confirm absence of innervation immunolabeling was used (see below).

**Vector construction and creation of transgenic lines.** Plasmid construction was based on the tol2/gateway zebrafish kit[61]. The p5E *pmyo6b* entry clone[9] was used to drive expression in hair cells, while the *SILL1* enhancer[12] in a p3E clone was used to drive expression in afferent neurons. For this study, we created pME-SypHy from a SypHy construct provided by Leon Lagnado[17], pME-jRCaMP1a-CAAX from Addgene clone #61532[62], and pME-Bongwoori from Addgene clone #63720[63]. The pME-GCaMP6s-caax clone has been previously described[13]. These clones were used along with the following tol2 kit gateway clones p5E-hsp70l (#222), p3E-2A-nlsmCherrypA (#766), p3E-polyA (#302), and pDest (#395), to create expression constructs: *myo6b:SypHy, myo6b:jRCaMP1a-CAAX, hsp70l:GCaMP6s-CAAX -SiLL1, and myo6b:Bongwoori-2A-nlsmCherry*. To generate stable transgenic fish lines Tg(-6myo6b:SypHy)$^{idc6Tg}$, Tg(-6myo6b:jRCaMP1a-caax)$^{idc7Tg}$, Tg(hsp70l:GCaMP6s-CAAX-SiLL1)$^{idc8Tg}$, and Tg(-6myo6b:Bongwoori-2A-nlsmCherry $^{idc9Tg}$), plasmid DNA and tol2 transposase mRNA were injected into zebrafish embryos as previously described[61].

**Immunohistochemistry and confocal imaging.** Immunohistochemistry was performed on whole larvae. Zebrafish larvae were fixed with 4% paraformaldehyde in PBS for 4.5–6 h at 4 °C. After 5 × 5 min washes in PBS + 0.01% Tween (PBST), and a 5-min wash in $H_2O$, larvae were permeabilized with ice cold acetone (at −20 °C) for 5 min. Larvae were then washed in $H_2O$ for 5 min, followed by a 5 × 5-min wash in PBST, and then blocked overnight with PBS buffer containing 2% goat serum and 1% bovine serum albumin (BSA). Primary antibodies were diluted (see below) in PBS buffer containing 1% BSA and larvae were incubated in the solution for 3 h at room temperature. After 5 × 5 min washes in PBST to remove the primary antibodies, diluted secondary antibodies (1:1000) coupled to Alexa 488 (#A21121, #A21131, #A21467), Alexa 568 (#A21133, #A11010), or Alexa 647 (#A21241, #A21242) (ThermoFisher, MA) were added in PBS buffer containing 1% BSA and incubated for 2 h at room temperature. After 5 × 5 min washes in PBST to remove the secondary antibodies, larvae were rinsed in $H_2O$ and mounted in Prolong gold (ThermoFisher Scientific, MA).

The following antibodies were used in this study:
anti-Vglut3 (1:1000, rabbit) labels hair-cell synaptic vesicles, a gift from Teresa Nicolson[60] anti-Vamp2 (1:500, rabbit, Genetex, #GTX132130) labels all hair-cell efferent presynapses
anti-Ribeye b (1:10,000, mouse IgG2a) labels hair-cell presynaptic ribbons, a gift from Teresa Nicolson[64]
anti-$Ca_V1.3a$ (1:1000, rabbit) labels hair-cell presynaptic $Ca^{2+}$ channels, a gift from Teresa Nicolson[64]
anti-pan-MAGUK antibody (1:500, mouse IgG1 NeuroMab, K28/86, #75–029) labels postsynaptic densities
anti-Ace-tubulin (1:5000, mouse IgG2b, Sigma, #T7451) labels afferent processes and hair cells
anti-TH (1:1000, mouse IgG2a, Vector labs, #VP-T489) labels dopaminergic efferents
anti-ChAT (1:500, goat, Millipore, #AB144P) labels cholinergic efferents.
In the zebrafish lateral line, in addition to afferent neurons, at least two types of efferent neurons innervate neuromasts: one is dopaminergic and the other is presumed to be cholinergic[8,21–23]. Both ChAT (cholinergic) and TH (dopaminergic) antibodies label efferent neurons innervating lateral-line

**Table 1 Pharmacological compounds utilized in the study**

| Mode of action | Drug | [Conc] | Incubation time (min) |
|---|---|---|---|
| L-type $Ca^{2+}$ channel $Ca_V1.3a$ antagonist | Isradipine | 10 μM | 10–20 |
| L-type $Ca^{2+}$ channel $Ca_V1.3a$ agonist | Bay K 8644 | 5 μM | 10–20 |
| Gap junction antagonist | FFA | 25 μM | 20–30 |
| Gap junction antagonist | 1-heptanol | 3 mM | 20–30 |
| AMPA/kainate receptor antagonist | DNQX | 10 μM | 10–20 |
| Disrupts hair-bundle tip links | BAPTA | 5 mM | 10 min and washed |
| Chloride channel antagonist | T16(inh)-A01 | 1 μM | 20–30 |

neuromasts (Supplementary Fig. 6a, c). Vamp2 antibody labels presynapses in both of these efferents (Fig. 4e, f and Supplementary Fig. 6a–d). Acetylated tubulin stains afferent processes (Fig. 4e, f and Supplementary Fig. 6g, h). Acetylated tubulin immunolabel colocalizes with the *neurod:GFP* transgene that has been shown previously to be expressed in afferent neurons innervating the lateral line[60] (Supplementary Fig. 6e, g, h). Based on these stainings, Vamp2 and Acetylated tubulin were used in combination to test for the presence or absence of efferent or afferent processes, respectively, in *neurog1a* morphants (Fig. 4e, f).

SypHy is a pH-sensitive GFP fused to the synaptic vesicle marker Synaptophysin[17]. To confirm SypHy localization in synaptic vesicles in hair cells, SypHy transgenic fish were immunostained with synaptic vesicle marker Vglut3 along with Ribeye b (Supplementary Fig. 4a–d). To confirm the presence of pre- and postsynaptic specializations in hair cells after SypHy imaging, fish were immunostained with presynaptic Ribeye b and postsynaptic MAGUK (Supplementary Fig. 4i, j).

Fixed samples were imaged on an inverted Zeiss LSM 780 laser-scanning confocal microscope with a 63× 1.4 NA oil objective lens; 488 nm was used for the excitation of Alexa 488, SypHy, or GFP; 546 nm for Alexa 568; and 647 nm for Alexa 647. For quantitative measurements, confocal imaging parameters, including gain, laser power, scan speed, dwell time, resolution, and zoom, were maintained between comparisons. The microscope parameters were adjusted using the brightest control specimen. For image analysis, maximal projections of z-stack confocal images were created and analyzed using ImageJ[65]. Images containing immunolabel were corrected for background using a rolling ball correction method.

For live images of transgenic fish in Supplementary Fig. 1, images were acquired on an upright Nikon C2 laser-scanning confocal microscope using a 60× 1.0 NA water objective lens. Appropriate solid-state lasers were used to image and excite EGFP, GCaMP6s, RGECO, SypHy, mCherry, jRCaMP1a, or Bongwoori.

**Pharmacology.** All drugs were prepared in extracellular solution with 0.1% DMSO (except no DMSO was used with BAPTA). For imaging experiments, the water-jet micropipette solution was also exchanged for an extracellular solution containing the drug. All drugs were purchased from Sigma-Aldrich except BAPTA, which was purchased from Thermo Fisher Scientific.

Drugs used include: isradipine, Bay K 8644, FFA, 1-heptanol, 6,7-dinitroquinoxaline-2,3-dione (DNQX), T16(inh)-A01, and 1,2-bis(2-Aminophenoxy) ethane-N, N, N′, N′-tetraacetic acid (BAPTA). Table 1 lists the concentrations and incubation durations of the drugs used in this study.

**Electron microscopy.** Larvae were prepared for electron microscopy as described previously[13]. Briefly, day 4 wild-type or *Tg(-6myo6b:Ribeye b-mCherry)idc3Tg* larvae were fixed in freshly prepared 4% paraformaldehyde and 2% glutaraldehyde (Electron Microscopy Sciences (EMS)) in 0.1 M phosphate buffer pH 7.4 for 30 min at room temperature, followed by a 2-h incubation at 4 °C. After fixation, larvae were washed with 0.1 M cacodylate buffer, and then fixed in 2% glutaraldehyde for 15 min, and washed with 0.1 M cacodylate buffer. Larvae then were placed in 1% osmium tetroxide in 0.1 M cacodylate buffer for 30 min and then were washed with 0.1 M cacodylate buffer, dehydrated in ethanol, including uranyl acetate in the 50% ethanol, followed by propylene oxide, and then were embedded in Epon. Transverse serial sections (~60 nm thin sections) were used to section through neuromasts. Most sections were placed on single slot grids coated with carbon and formvar (EMS), and then sections were stained with uranyl acetate and lead citrate. Samples were imaged on a JEOL JEM-2100 electron microscope (JEOL Inc.).

Gap junctions were identified by established criteria for standard TEM: two cell membranes aligned parallel and relatively straight, with an intercellular gap of 2–3 nm (~5 nm including the outer stained leaflets of the two cell membranes)[25,27,66,67]. The appearance of published high magnifications of the gap junction substructure is variable, and dependent on fixation, embedding, sectioning, staining, and imaging[66,67]. We examined all hair cells and supporting cells in the sections, searching for gap junctions between the cells along their sides or bases. The gap junctions that we identified in the neuromasts appear to be identical in fine substructure to those found between supporting cells in the vestibular or auditory epithelium of vertebrates[25–27,68]. We did not find any gap junctions between hair cells and supporting cells, nor did we find any gap junctions among the afferent and efferent nerve processes. Gap junctions were relatively rare—for example, in one experiment where we examined 29 sections from 4 neuromasts

from two wild-type larvae we found 7 gap junctions; for 2 out of these 7, the gap junctions were found in 2 and 3 serial sections, respectively.

**Sample preparation and stimulation for functional imaging.** To prepare larvae at 2–6 dpf for functional imaging, individual larvae were first anesthetized with tricaine (0.03% ethyl 3-aminobenzoate methanesulfonate salt, Sigma). For older larvae at 13 dpf (Fig. 2h), larvae were incubated with tricaine and decapitated with a razor blade. Whole or decapitated larvae were then pinned onto a Sylgard-filled recording chamber. To suppress movement in intact larvae, alpha-bungarotoxin (125 μM, Tocris) was injected into the heart. Larvae were then rinsed with extracellular imaging solution (in mM: 140 NaCl, 2 KCl, 2 $CaCl_2$, 1 $MgCl_2$, and 10 HEPES, pH 7.3, OSM 310±10) without tricaine and allowed to recover. Stimulation of neuromast hair cells was accomplished using a fluid jet. The fluid jet consisted of a pressure clamp (HSPC-1, ALA Scientific) attached to a glass pipette (inner tip diameter ~30–50 μm). The glass pipette was filled with extracellular solution and used to mechanically stimulate the apical bundles of hair cells along the anterior–posterior axis of the fish[9]. We used the fluid jet to stimulate the two polarities of hair cells by applying an anterior or posterior directed stimulus separately (Supplementary Fig. 3a–c, e–g). Alternatively, we used a 2-s long 5 Hz alternating anterior–posterior directed stimulus in the form of a square wave to stimulate all hair cells simultaneously, within the same trial (Supplementary Fig. 3a, d, e, h). We obtained similar results with both stimulus paradigms. Apical hair-bundle deflection was monitored by measuring the displacement distance of tips of the bundles, the kinocilia. For a 2-s deflection (step or 2-s 5 Hz), we found that when the kinocilial tips were deflected more than 5 μm, the magnitude of the apical and presynaptic GCaMP6s $Ca^{2+}$ signals were saturated. Within 1–5 μm kinocilial deflection distances, $Ca^{2+}$ signals increased with increasing deflection distance and the $Ca^{2+}$ signals were not saturated. Intense bundle deflections, that moved kinocilial tips more than 10 μm commonly induced movement artifacts, and were damaging to hair bundles. Consistent with damage, after intense 10 μm bundle deflections, subsequent $Ca^{2+}$ responses were dramatically reduced. A 5-μm deflection distance was used in the majority of our experiments to achieve maximal $Ca^{2+}$ signals. For Fig. 1, and to measure the correlation between apical and presynaptic $Ca^{2+}$ signals, a kinocilial displacement distance of 2 μm was used to ensure that apical and presynaptic responses $Ca^{2+}$ remained unsaturated. For detection of afferent $Ca^{2+}$ signals, a shorter 200 ms stimulus (Fig. 3e) provided a more accurate postsynaptic readout compared to 2 s stimulus (Fig. 3l). During longer stimuli, postsynaptic $Ca^{2+}$ signals spread from their sites of origin—adjacent to ribbons—to the entire afferent process. During longer stimuli, active postsynaptic sites were determined by stimulus onset. To coordinate stimulation with functional image acquisition, the pressure clamp was driven by a voltage-step command during the acquisition. An outgoing voltage signal from the imaging software (Prairie view or Nikon Elements) was used to coordinate image acquisition with the pressure clamp stimulus.

For functional imaging pre- and post-decapitation (Fig. 4a–d), we first assayed the presynaptic $Ca^{2+}$ responses before decapitation of the larval zebrafish. Then we decapitated the fish with a razor blade and restabilized the remaining fish body with pins. Within 20 min, we re-assayed the presynaptic $Ca^{2+}$ responses by applying the same stimulation.

For GCaMP6s experiments using high $K^+$ application to depolarize hair cells, the fluid jet was used to mechanically stimulate hair cells and identify active hair cells. Then animals were treated with BAPTA to cleave the extracellular linkages among stereocilia (tip links[69]) required to gate the mechanosensitive ions channels (see additional controls below), and abolish mechanosensitive function. After BAPTA treatment, the fluid jet was no longer able to mechanically generate detectable presynaptic $Ca^{2+}$ responses in hair cells (Fig. 6g). The solution inside the fluid jet was then replaced with 1 M KCl and positioned 100 μm from the neuromast. An anterior-directed stream of $K^+$ was applied for 6 s during GCaMP6s acquisition to deliver high $K^+$; 1 M $K^+$ was required to penetrate the skin and supporting cells in order to activate the hair cells during the stimulus time-window in our intact preparation. For Bongwoori voltage imaging experiments, BAPTA treatment and the high $K^+$ stimulus was executed in a similar manner.

**Verification of genetically encoded indicator specificity.** To confirm that the apical, mechanosensitive $Ca^{2+}$ responses detected using *myo6b:GCaMP6s-CAAX*

were not due to motion artifacts, but were due to activities of mechanosensitive ion channels present in mechanosensory bundles, we applied BAPTA to abolish mechanosensitivity. This eliminated mechanosensitive $Ca^{2+}$ responses at the apex as well as presynaptic $Ca^{2+}$ responses at the base (Supplementary Fig. 2a–e). To confirm that presynaptic $Ca^{2+}$ responses at the base of the hair cell were dependent on $Ca_V1.3$, we applied the L-type $Ca^{2+}$ channel antagonist isradipine. Isradipine eliminated all presynaptic $Ca^{2+}$ responses at the base of the hair cell, with no impact on apical mechanosensitive signals (Supplementary Fig. 2f–j).

To test the specificity of the afferent $Ca^{2+}$ responses detected using *hsp70l: GCaMP6s-CAAX-SiLL1*, we applied DNQX, an AMPA receptor antagonist as recent work has shown that $Ca^{2+}$-permeable AMPA receptors are required for postsynaptic $Ca^{2+}$ responses at hair-cell synapses[70]. After DNQX application, the number of hair cells associated with postsynaptic $Ca^{2+}$ activity was significantly decreased (Fig. 3f).

SypHy is a pH-sensitive GFP localized to synaptic vesicles. Inside synaptic vesicles SypHy is acidified and its fluorescence is quenched, but upon fusion with the plasma membrane, SypHy becomes deacidified and increases in fluorescence. To confirm that SypHy is properly acidified and poised to detect vesicle fusion in all cells, the extracellular solution was exchanged with a similar solution containing 40 mM $NH_4Cl$ (40 mM NaCl was removed to maintain the osmolarity) to deacidify and unquench SypHy. SypHy baseline signals were imaged with an upright Nikon C2 laser-scanning confocal microscope using a 60× 1.0 NA water objective lens before and after a 10–20 min 40 mM $NH_4Cl$ application. ImageJ[65] was used to subtract background and quantify intensity changes before and after de-acidification. Relative to the baseline SypHy signal, all hair cells showed an increased SypHy signal after $NH_4^+$ application (Supplementary Fig. 4e–g), indicating that SypHy was properly acidified in all hair cells.

### Functional imaging.

For imaging $Ca^{2+}$-dependent mechanosensation, presynaptic $Ca^{2+}$, cytosolic $Ca^{2+}$, afferent $Ca^{2+}$, SypHy vesicle fusion, and Bongwoori voltage changes, we used two microscope systems: a Bruker Swept-field confocal system, and an upright motorized Nikon ECLIPSE Ni-E microscope working in either widefield and confocal imaging modes. The Bruker Swept-field system was used for all hair-cell SypHy, RGECO1, jRCaMP1a, Bongwoori imaging, and the majority of the GCaMP6s presynaptic $Ca^{2+}$ imaging, as well as all afferent GCaMP6s imaging (Figs. 1–6, Supplementary Figs. 2–7). The Nikon widefield and confocal system was used for presynaptic GCaMP6s imaging before and after damage (Fig. 7).

The Bruker Swept-field confocal system was equipped with a Rolera EM-C2 CCD camera (QImaging) and a Nikon CFI Fluor 60× 1.0 NA water immersion objective. We used a dual band-pass 488/561 nm filter set (59904-ET, Chroma). The system was controlled by Prairie view (Bruker Corporation). Single plane acquisitions were taken for all hair-cell SypHy, RGECO1, jRCaMP1a, and Bongwoori imaging, at a frame rate of 10 Hz with 2 × 2 binning. SypHy imaging data were acquired in three separate single planes 2 μm apart that were combined for analysis. Simultaneous imaging of $Ca^{2+}$ activity at all pre- or postsynaptic sites using GCaMP6s in hair cells or afferent processes across the Z-axis was accomplished by using a piezoelectric motor (PICMA P-882.11-888.11 series, PI instruments) attached to the objective to allow rapid imaging in five planes along the Z-axis with 2 μm intervals, at a 50-Hz frame rate yielding a 10-Hz volume rate. Five plane Z-stacks were projected into one plane for further image processing and quantification. For analysis of ribbon-localized $Ca^{2+}$ signals, only the plane containing the ribbons was used for analysis (Fig. 1h, i).

Dual-color RGECO1 and SypHy or GCaMP6s imaging was accomplished using a Dual-View beam splitter (Photometrics) with the following filters: dichroic 565; green emission 520/30; red emission 630/50 (Chroma) to spectrally separate the indicators, and enable dual imaging of green and red signals. RGECO1 and SypHy or GCaMP6s imaging data were acquired sequentially. A single, central RGECO1 imaging plane was acquired first (Fig. 3h, Supplementary Fig. 5b). Next, three separate SypHy and GCaMP6s imaging planes were acquired 2 μm apart (Fig. 3i and Supplementary Fig. 5c) with a beam splitter. The latter three planes were projected for analysis.

### Laser damage experiments.

The Nikon ECLIPSE Ni-E microscope was used for the laser damage experiments. For these experiments, hair-cell GCaMP6s pre-synaptic $Ca^{2+}$ measurements were detected using either a widefield ORCA-D2 camera (Hamamastu Photonics) (Fig. 7) or C2 confocal PMTs controlled using Elements software (Nikon Instruments Inc.). For widefield GCaMP6s imaging, the microscope was equipped with the following filter set: excitation: 480/30 nm and emission: 535/40 nm, with 2 × 2 binning. For the C2 confocal system, GCaMP6s was excited using a 488-nm solid state laser. For both imaging methods, a 60× 1.0 NA CFI Fluor water-immersion objective was used, and functional $Ca^{2+}$ imaging was performed at 6–10 Hz. After stimulation to determine which hair cells were active, in Nikon Elements, a focused point ROI was placed on the nucleus of a single active cell. Damage to hair cells was performed by a 405-nm laser focused on the point ROI at 100% laser power, using a scan speed of 8 ms, for 1.5–2 s. This damage intensity was chosen because it is at the threshold of inactivation, but the damage did not overtly alter any cellular structures. For example, a 405-nm laser damage duration of 1.0 s was not sufficient to inactivate a hair cell. We defined successful laser damage as a damaged cell that was still present, but was no longer

presynaptically responsive (example, cell 1 in Fig. 7b, e). After damage to the active hair cell, the organ was restimulated and assessed 10–30 min after laser damage.

### $K^+$ concentration measurements.

Asante Potassium Green-2 (APG-2, Kd = 18 mM, TEFLabs) dye was employed to measure the relative $[K^+]_{in}$ levels in hair cells and supporting cells. Unlike many vital dyes, APG-2 does not enter hair cells through mechanosensitive channels. To label hair cells with APG-2, we injected 100 μM APG-2 into the heart, along with alpha-bungarotoxin used to paralyze the larvae. Larvae were then bathed in extracellular solution for 30 min to allow the dye to label hair cells before imaging. Further incubation time did not result in additional dye labeling. APG-2 was imaged on a Nikon C2 confocal microscope (see above) using a 488-nm laser. For hair-cell $Ca^{2+}$ imaging using jRCaMP1a and subsequent APG-2 labeling, larvae were prepared for presynaptic $Ca^{2+}$ imaging as described above for GCaMP6s using our swept-field confocal system. The jRCaMP1a line resulted in dramatically less spectral bleedthrough into the green channel compared to RGECO1. Eliminating bleedthrough was especially important when performing dual-color imaging with the relatively dim APG-2 dye. After jRCaMP1a functional imaging, larvae were then heart injected with 100 μM APG-2 and reimaged after 30 min on the Nikon C2 confocal microscope. ImageJ[65] was used to subtract background and quantify APG-2 intensity.

### Functional imaging registration and processing.

The raw images were processed using a custom program with a user-friendly GUI interface in MATLAB R2014 (MathWorks). We first removed the first 10 images to reduce the effect of photobleaching. Then the raw images were registered to reduce movement artifacts by applying efficient subpixel image registration based on cross-correlation[71]. The procedure to obtain and overlay the spatial signal distribution as heat maps has been described previously[14]. Briefly, we first computed the baseline image (or reference image) by averaging the images over the whole pre-stimulus period, pixel by pixel (approximately 20 frames). Then the baseline image ($F_0$) was subtracted from each image acquired, and then each image was divided by $F_0$ to generate images that represent the relative change in fluorescent signal from baseline or $\Delta F/F_0$. To better visualize the fluorescence intensity changes during stimulation, $\Delta F/F_0$ signal images over the stimulus period were averaged, scaled, and encoded by color maps with blue indicating a decrease in signal intensity in hair-cell Bongwoori imaging, and red indicating an increase in signal intensity for all the other indicators. For each indicator (and for each microscope), we determined the noise floor —a value above which we could reliably discern signal from noise—and used this value consistently in our figures. The noise floor helped establish the starting value for our $\Delta F/F_0$ color maps. The color maps were then superimposed into the averaged baseline raw grayscale images, in order to easily visualize the spatial fluorescence intensity changes in all the hair cells or afferent processes within a neuromast during stimulation. For the data presented in the figures, the average relative change in fluorescent signal during the entire stimulation period is displayed as "stim".

### Functional imaging quantification.

For SypHy vesicle fusion, presynaptic GCaMP6s and jRCaMP1a, cytosolic RGECO1 and afferent GCaMP6s $Ca^{2+}$ measurements, a circular ROI with a diameter ~3 μm (12 pixels with 268 nm per pixel) was placed on each hair cell within a neuromast. For Bongwoori, the voltage sensor, a circular ROI with a diameter ~5 μm was used to better incorporate the cell membrane, where the voltage changes were restricted. For hair-bundle and ribbon-localized presynaptic GCaMP6s measurements, a circular ROI with an ~1.5 μm diameter was placed on the center of an individual bundle or ribbon. Ribbon location was determined by either simultaneous (using a dual-view splitter, see above), or subsequent image capture of Ribeye-mCherry labeled ribbons. Ribbon-localized signals (Fig. 1h) were measured in only the plane of the ribbon, rather than a z-stack projection containing all ribbons. After selecting an ROI, we computed and plotted the mean intensity ($\Delta F/F_0$) within each ROI during the recording period. The temporal signal plots were smoothed by 3-point data rolling to reduce the noise-induced oscillation while maximizing the original timing accuracy. The signal magnitude was defined as the peak value of intensity change upon stimulation.

### Statistical analysis.

All data were plotted with Prism 7 (Graphpad). Values in the text and data with error bars on graphs and in text are expressed as mean ± SEM. All experiments were performed on a minimum of three animals, five neuromasts, and on two independent days. For each neuromast, we sampled approximately 8 (day 2), 16 (day 5), and up to 20 (day 13) hair cells (Fig. 2h, right Y-axis). These numbers were adequate to provide statistical power to avoid both Type I and Type II error. No animals or samples were excluded from our analyses unless control experiments failed—in these cases all samples were excluded. No randomization or blinding was used for our animal studies. Where appropriate, datasets were confirmed for normality using a D'Agostino-Pearson omnibus normality test and for equal variances using an $F$ test to compare variances. Welch's correction was used if samples did not have equal variances. Statistical significance between two conditions was determined by either paired or unpaired, two-tailed Student's $t$-tests, Mann–Whitney test, or Wilcoxon matched-pairs rank test as appropriate. For

comparison of multiple conditions, a one-way ANOVA with a Tukey's test to correct for multiple comparisons was used.

**Code availability**. Matlab R2014 was used to process functional imaging data and the code is available upon request.

**Data availability**. The data that support the findings of this study are available from the corresponding author upon reasonable request.

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

## Acknowledgements

This work was supported by National Institute on Deafness and Other Communication Disorders Intramural Research Program Grant 1ZIADC000085-01 to K.S.K. and ZICDC000081 to R.S.P. and Y.-X.W. We would like to thank Lisa Cunningham, Catherine Drerup, Catherine Weisz, Elyssa Monzack, Doris Wu, Lavinia Sheets, Alex Chesler, and Daria Lukasz for their support and thoughtful comments on the manuscript.

## Author contributions

Q.Z. and K.K. conceived, designed, performed the experiments, analyzed the data, and wrote the manuscript. X.J.H., S.L., and H.C.W., designed, performed the experiments, and analyzed the data. A.B., R.S.P., and Y.-X.W. performed the experiments.

## Additional information

**Competing interests:** The authors declare no competing interests.

