## [Peer Review File · Nature Communications]

Reviewers' comments:

Reviewer #1 (Remarks to the Author):

This is an interesting manuscript that suggests that all mechanosensitive hair cells are not synaptically engaged. The fundamental finding is novel and well justified and documented. It posits that hair cells can become competent quickly after hair cell loss. It further posits that supporting cells are involved in maintaining intracellular potassium concentrations in hair cells and that this regulates synaptic competency. Data is of a high quality and presentation is reasonably clear. There are both major and minor issues with this manuscript as it stands. In general, the paper is descriptive of a phenomenon, but provides no realistic hypothesis or test of hypothesis for either how it happens or what the physiological relevance might be. In addition, the basic finding should be discussed in light of function of the lateral line system where it is documented and less in how it might relate to other systems where it may or may not be relevant. Considerable data is presented to justify the basic finding, none is presented to support a mechanism or function. Detailed concerns include:

Major Issues:

- 1- Authors argue that not having all hair cells synaptically active alters the system transfer function to the brain; however, it is unclear that afferent fiber firing rates are impacted. What percentage of fibers are functional? I would assume all fibers are functional, despite all synapses not being functional and the number of hair cells driving the firing rate is likely irrelevant. Without some idea of the input/output function a case regarding plasticity is very difficult to make.
- 2- Are calcium channels present at every ribbon? Might there be a population of immature hair cells whose calcium channels have not yet localized to the presynaptic zone? Or might there be a population of calcium channels that are inactivated because the hair cells are depolarized?
- 3- Related to 2, what is the expected mechanism by which hair cell depolarization does not activate calcium channels? Or are the authors suggesting that MET channel activation does not depolarize the hair cells? Furthermore, how does lower internal K lead to synapse function? There is no clear discussion or path to defining or testing a feasible hypothesis for the presented data. A lot of effort is spent on saying that there are hair cells that are not synaptically active because there is no elevated calcium at the synaptic regions but not much effort to explain how this can happen or what the relevance might be.
- 4- It is unclear how gap junctions can lower intracellular potassium in hair cells. Is potassium entering via mechanotransduction channels and leaving through voltage-gated channels to be sucked up by supporting cells? How does it get into supporting cells, transported? Did you investigate K levels in supporting cells or voltage of supporting cells? Are cells with lower potassium depolarized because of the shift in Nernst potential? Would depolarization make the cells more or less responsive to hair bundle deflection? This could be argued either way as input resistance is likely lower as is driving force and calcium channel inactivation likely more prevalent, all contributing to reducing sensitivity of the hair cells. Alternatively, a smaller receptor potential might be more effective if the cell's resting potential overlaps with the calcium channel activation curve.

Other comments:

- 1- Abstract and intro: authors spend more time than necessary trying to justify the model system. This seems unnecessary and somewhat irrelevant to their story. Lateral line is not a model for mammalian cochlea and shouldn't be presented that way, it is an important system on its own.
- 2- Please define suprathreshold mechanical stimulation, term is vague and so uninformative.
- 3- Please be consistent when discussing low resting potassium that you are referring to internal K and not extracellular K.
- 4- Too much time spent on justifying a model system, particularly as representative of other inner ear organs as this is simply not true at most levels. Should simply discuss as mechano detector in context with what the lateral line actually does (this is really overlooked). Line 66 amicable is a strange choice of words.
- 5- Line 86, what does 'complete mechanotransduction' mean?
- 6- Line 92 What does redundant mean? How are silent cells redundant? Do they not serve to replace damaged cells, is this not an important role that might be specialized to a lateral line system where rapid replacement is necessary for survival, i.e. regeneration would be too slow.
- 7- Line 109 Enriched typically means higher concentration per unit area, is this what the authors mean or are they saying that the surface area to volume ratio is greater in the hair bundle so the apparent level of indicator is higher?
- 8- Lines 119-122 is there any attempt to quantify the level of calcium entry per bundle as a means of assessing relative effects on receptor potential? Are the signals all comparable or do some get brighter or stay brighter longer than others?
- 9- Is there any difference in baseline levels of calcium either in the stereocilia or basolaterally that might support an argument for cells being more depolarized or having more active MET?
- 10- Is there any correlation between the level of calcium signal apically to that basolaterally? Is there a threshold level of MET that needs to happen to trigger a basolateral effect? Line 152 suggests not but there is no quantitation presented or description of how this was measured. This needs to be included.
- 11- Do all hair cells have calcium channels clustered at synapses. This needs to be definitively answered for any interpretation of mechanism or function.
- 12- Line 174 How is a saturating mechanical stimulation defined?
- 13- What is the innervation pattern of hair cells with respect to functional synapses? Are all afferent fibers innervated by some percentage of functional synapses or are there afferent fibers that are not mechanically sensitive? This is a critical question as it relates to both physiological relevance as well as potential mechanisms.
- 14- Line 311, what does mainly detected mean? Should this be quantified. Were there synapses that did not have major calcium responses that still showed vesicle fusion?
- 15- Why not simply depolarize the entire neuromast with potassium to see if all hair cells could be activated.
- 16- The ablation experiment is interesting and valuable as it perhaps points to the function of having nonresponsive hair cells that can rapidly assume function. Were there any behavioral tests done to show that in fact there was compensation at the systems level? Were there any behavioural tests to show that there was a functional deficit when a hair cell was ablated?

- 17- The ten-minute time frame is interesting but it is unclear that it precludes a change in protein expression or distribution, in particular as it relates to the calcium channels. Channels may be expressed but not inserted and a shift can occur in this time frame.
- 18- The efferent experiments were done well and relatively thoroughly but it is unclear that this was not a straw man type of set of experiments as the means by which the efferent system could produce this result is unclear. Are innervation patterns such that individual fibers would be selectively diminishing activity in a population of hair cells? Is there any correlation between efferent patterns and patterns of hair cell functionality? It is somewhat unclear as to why this direction was chosen for investigation.
- 19- Does the loss of efferent/afferent prevent the recovery of functional hair cells after hair cell ablation? That is could either of these pathways be involved in detection or replacement of lost hair cells as compared to the establishment of the hair cell functional pattern?
- 20- A single TEM image is not really appropriate for defining an expression pattern within the neuromast. The distribution and expression level would be more appropriate. Also, how close do membranes need to be to guarantee that it represents a gap junction? Shouldn't they be directly abutting each other and not simply closer together?
- 21- What is the Kd of the potassium sensor? Was this calibrated with fluorescent output to get an idea of what the concentration difference is for better estimates of Nernst potential changes?
- 22- Line 446 need to be clear as to intra or extra cellular potassium levels.
- 23- The discussion is weak as it includes a lot of vagueness about potential mechanisms and about potential function. It also attempts to discuss other hair cell organs which may or may not be relevant but it excludes discussing potential functional relevance to the lateral line organ. There are no clear feasible mechanisms discussed nor tested. The functional relevance is speculative and not well founded.

Reviewer #2 (Remarks to the Author):

This is a very clearly presented study of sensory transduction in lateral line organs of the zebrafish. Using in vivo imaging of fluorescent reporters and genetic manipulations, the authors report several exciting new findings. Principally, that only a few hair cells within a neuromast are able to excite their synaptically associated afferent despite experiencing robust calcium influx into the bundle following deflection. This suggests that some hair cells are held in reserve – a concept supported by the observation that damaging a functionally coupled hair cell often triggered a “latent” hair cell to become functionally coupled. This plasticity was remarkably rapid, occurring within 10 minutes after injury. These studies are elegant and well supported by the many aspects of transduction that were analyzed through highly complementary approaches.

The second part of the study investigated the mechanisms that regulate the latent nature of the functionally uncoupled hair cells. The authors provide evidence that this is not controlled by either efferents or afferents. Again, very convincing manipulations. They then explore whether supporting cell gap junctions are responsible, showing that gap junctions are present between supporting cells and that blocking these channels pharmacologically

reduces presynaptic calcium responses and synaptic coupling. Further, the authors analyze K⁺ levels in hair cells using a K⁺ sensitive dye and find that functionally coupled cells have lower resting K⁺ levels and that inhibiting gap junctions elevated K⁺ levels. Together, these findings suggest that gap junctional coupling among supporting cells maintains functional coupling by regulating intracellular (and perhaps extracellular) K⁺ levels.

Overall, this study provides new insight into the coordination among hair cells within neuromasts and suggests that both plasticity and functional redundancy are used to maintain organ function. Enthusiasm is high for the first part of the story, which I would support for publication without the second part. The second part of the study involving gap junctions and K⁺ control is more preliminary and, while the results suggest that there may be some involvement of supporting cells, the studies are not definitive, due to the weak selectivity of the pharmacological agents used and the limited manipulations performed.

Major comments

1. Selective inhibition of gap junctions is difficult to achieve pharmacologically. FFA and heptanol have many other effects (particularly on chloride channels). Genetic manipulation of gap junctions in supporting cells would provide much stronger evidence for their involvement.

2. Missing from the study is any assessment of hair cell membrane potential. Knowledge about these changes will be crucial to provide further evidence for the K⁺ hypothesis. For example, it would be nice to know if latent hair cells experience the same amount of depolarization in response to hair bundle deflection. Is it possible that these cells have lower membrane resistance? If so, one might assume that inhibiting K⁺ channels would promote recruitment of latent cells. If genetically encoded voltage sensors work in this system it would provide a nice complement to the calcium imaging, as it would help define how depolarization-excitation coupling is rapidly altered. In my mind, it would be sufficient to just speculate about some potential mechanisms in the discussion.

3. The authors suggest that supporting cell gap junctions somehow create local K⁺ gradients that differentially alter the excitability of hair cells. This is an unconventional view of the role of gap junctions, as gap junctions are typically thought to normalize ion gradients. One would expect that they would help to buffer K⁺, dissipating transients generated by active cells through the glial syncytium, rather than produce persistent local increases. It would be helpful if the authors were to provide a model to explain how they think supporting cells influence hair cell excitation by this mechanism.

4. In the experiments that have been performed, the functional consequence of the different intracellular K⁺ levels has not been established (it is not clear how much K⁺ varies between these cells or how inhibiting supporting cell gap junctions would lead to changes in intracellular K⁺ in hair cells). In addition, the studies in which extracellular K⁺ was changed are complicated by the fact that this manipulation will also induce depolarization and perhaps inactivation of calcium channels.

Minor

1. In Fig. 1b4 – the line colors do not match the legend
2. The pseudocoloring in Fig. 1 should be standardized for the two experiments to facilitate comparisons (and start from 0).
2. Reword: While we observed that fewer hair cells with vesicle fusion during a 50 Hz stimulus, for all other stimuli tested, a similar percentage of hair cells per neuromast had vesicle fusion (Figure 2c, one-way ANOVA, $df = 30$).

Subject: Response to reviewers comments

Manuscript ID: NCOMMS-17-20458-T

Manuscript Title: Gap junction networks facilitate presynaptic activity in sensory hair cells

Authors: Qiuxiang Zhang, Suna Li, Hui-Tung C. Wong, Xinyi J. He, Alisha Beirl, Ronald S. Petralia, Ya-Xian Wang, and Katie S. Kindt

We sincerely thank the reviewers for their valuable comments and constructive suggestions to improve the manuscript. In this revision, we have done our best to address all of the reviewers' comments.

Based on your comments, the main changes are as follows:

1. **Confirmed the presence of $Ca_v1.3$ channels at synapses**
2. **Used high K^+ to demonstrate that hair-cell depolarization is unable to activate $Ca_v1.3$ channels in silent cells**
3. **We created a new voltage indicator line to demonstrate that all hair cells are depolarized by high K^+**
4. **Our discussion now clearly outlines how gap junctions could regulate intracellular K^+ to regulate $Ca_v1.3$ channel activity in active and silent neuromast hair cells**
5. **The manuscript is more concise and now fits into the Nature Communications 5,000 word limit**

Following is an additional summary of our response to the reviewers' questions/concerns. For your convenience, we cite each of the reviewers' comments before our response initiated with "**Response**".

Line numbers match unmarked document.

Response to Reviewer 1

Reviewer's concern (Main Comments 1): "Authors argue that not having all hair cells synaptically active alters the system transfer function to the brain; however, it is unclear that afferent fiber firing rates are impacted. What percentage of fibers are functional? I would assume all fibers are functional, despite all synapses not being functional and the number of hair cells driving the firing rate is likely irrelevant. Without some idea of the input/output function a case regarding plasticity is very difficult to make."

Response:

We agree we have not shown how afferent fiber spike rate is impacted by having only a subset of active hair cells. We have added additional information to the introduction and discussion to clarify the wiring present in neuromasts.

Introduction: Page 3, Line 58-66... *"The anatomical composition of primary, posterior lateral-line neuromasts is well-defined. In each neuromast there are two populations of hair-cells with bundles polarized to respond to stimuli directed in either an anterior or posterior direction[1, 2]. At the base of the neuromast, each hair cell has on average 3 presynapses or 'ribbons' that tether synaptic vesicles at the active zone near $Ca_v1.3$ channels[3]. Postsynaptically, each neuromast organ is innervated by multiple afferent neurons. **Each afferent neuron contacts nearly all hair cells of the***

same polarity, and hair cells are contacted by more than one afferent neuron[4]. Overall this anatomy describes a sensory system stacked with anatomical redundancy at many levels—multiple hair cells per polarity, synapses per hair cell, and postsynaptic afferents contacts per hair cell.”

Discussion: Page 19, Lines 413-417... *“Previously, three levels of anatomical redundancy have been described within neuromasts. First, each of the two hair-cell populations (responding to either an anterior or posterior directed stimulus[2]) are represented by multiple hair cells. Second, each hair cell has approximately three synapses per hair cell[3] and third, each hair cell is innervated by more than one afferent neuron[4]”*

The anatomy of neuromasts suggests that this organ system is wired to maximize detection even if not all hair cells are not transmitting and does suggests there are spikes in all afferent neurons. In the lab, we have done preliminary experiments (for another project) by imaging (calcium) activity in single afferent fibers and we do indeed see activity in all fibers. Because we have now clarified the anatomy (above) and the resulting plasticity (below), we feel that this data is not essential for this particular study.

With regards to plasticity, it is difficult to know how afferent spike rate is changed after laser damage. Previously in our laser damage experiments, we found that after damage, the new active cell did not always have the same polarity as the damaged cell. Therefore, the plasticity does not replace an equivalent cell and this backup strategy does not use logic based on wiring. Because of this non-equivalent replacement strategy, afferent spike rate could remain the same, or even decrease after damage. We have tried to clarify this in the manuscript:

Results: Pages 10-11, Lines 221-226... *“Of the two polarities of hair cells within neuromast organs (responsive to either an anterior or posterior directed stimulus), after damage, the new active cell could arise with the same or opposite polarity of the damaged cell (same direction: 57 %, opposite direction: 43 %). Because the damaged cell was not always replaced with an equivalent hair cell, this indicates that the resulting plasticity is stochastic, and functions broadly to maintain a proportion of active cells within the population.”*

Discussion: Pages 19, Lines 417-421... *“Surprisingly, unlike these anatomical redundancies, after laser damage, presynaptic unsilencing does not always replace an equivalent hair cell. After damage and loss of a hair cell from one population, the unsilenced cell could arise from either of the two hair-cell populations. This suggests that synaptic activity states may be set up or transferred stochastically without regard to anatomy.”*

Reviewer’s concern (Main Comments 2): *“Are calcium channels present at every ribbon? Might there be a population of immature hair cells whose calcium channels have not yet localized to the presynaptic zone? Or might there be a population of calcium channels that are inactivated because the hair cells are depolarized?”*

Response: Great suggestion. In our revision, we have stained neuromast hair cells with a Ca_v1.3 antibody and found that clusters of Ca_v1.3 are present at nearly all ribbons.

Figure 5a-b: Example immunostaining of Ca_v1.3 and ribbons (Ribeye b). Quantification shows 93% of ribbons per neuromast co-label with Ca_v1.3.

Results: Page 11, Lines 232-235... *“This suggests that presynaptic Ca²⁺ channels are present in all neuromast hair cells. To confirm this, we used immunohistochemistry and observed that ~93 % (Fig. 5b) of presynapses within a neuromast are associated with clusters of the voltage-gated Ca²⁺ channel Ca_v1.3 (Example, fig. 5a-a”).*”

See depolarization explanation below.

Reviewer’s concern (Main Comments 3): *“Related to 2, what is the expected mechanism by which hair cell depolarization does not activate calcium channels? Or are the authors suggesting that MET channel activation does not depolarize the hair cells? Furthermore, how does lower internal K lead to synapse function? There is no clear discussion or path to defining or testing a feasible hypothesis for the presented data. A lot of effort is spent on saying that there are hair cells that are not synaptically active because there is no elevated calcium at the synaptic regions but not much effort to explain how this can happen or what the relevance might be.”*

“Related to 2, what is the expected mechanism by which hair cell depolarization does not activate calcium channels? Or are the authors suggesting that MET channel activation does not depolarize the hair cells?”

Response: We have now added data to show that while it is likely that MET activation depolarizes all hair-cells, depolarization is unable to activate Ca_v1.3 in all cells.

Figure 5c-c’: We used a fluid-jet to mechanically activate hair cells (using the GCaMP6s calcium indicator) to identify active cells (c’). Then we applied BAPTA to eliminate MET function (c’). Lastly we used high K⁺ to depolarize hair cells (c’’). We found that the same set of cells is activated by the fluid-jet and high K⁺. This suggests that depolarization does not activate Ca_v1.3 channels in all cells.

Results: Page 11-12; Lines 243-249... *“For these experiments, we first we used a mechanical stimulus and GCaMP6s to identify active hair cells (Fig. 5c-c’). Then we applied BAPTA to eliminate all mechanically evoked responses (Fig. 5c’’, and Supplementary fig. 2a-a’’, b-b’). Then we applied high K⁺ via the fluid jet to depolarize all hair cells within a neuromast. Upon high K⁺ application, we observed the same overall pattern of presynaptic Ca²⁺ signals as using a mechanical stimuli (Example fig. 5c’’’, n = 6 neuromasts). This indicates either high K⁺ induced depolarization is unable to activate Ca_v1.3 channels in silent cells, or high K⁺ is unable to depolarize all hair cells.”*

Figure 5d-d’: We also repeated the high K⁺ experiments but instead of GCaMP6s we expressed Bongwoori, a voltage indicator in hair cells. With Bongwoori we show that all hair cells are depolarized by high K⁺ (examples, d-d’’’ and e-e’’’). Our GCaMP6 and Bongwoori data together suggest that hair-cell depolarization is unable to activate Ca_v1.3 in all cells.

Results: Page 12; Lines 248-256... *“This indicates either high K⁺ induced depolarization is unable to activate Ca_v1.3 channels in silent cells, or high K⁺ is unable to depolarize all hair cells. To distinguish between these scenarios, we created a transgenic line expressing the genetically-encoded voltage indicator Bongwoori in hair cells (Supplementary fig. 1c) to measure membrane voltage changes rather than presynaptic Ca²⁺ signals. Using Bongwoori, we found that high K⁺*

application robustly depolarized all hair cells within a neuromast (Examples fig. 5d-e''', n = 7 neuromasts). Overall using high K^+ application, along with hair-cell Ca^{2+} and voltage imaging, our data suggest that hair-cell depolarization is unable to activate $Ca_v1.3$ channels in silent cells."

Furthermore, how does lower internal K lead to synapse function? There is no clear discussion or path to defining or testing a feasible hypothesis for the presented data. A lot of effort is spent on saying that there are hair cells that are not synaptically active because there is no elevated calcium at the synaptic regions but not much effort to explain how this can happen or what the relevance might be."

Response: In this revision, we had added additional information regarding how lower internal K^+ could impact synapse function. We have also discussed how this might happen in the discussion, as well as the biological relevance. Although we would like to know more about the exact mechanisms involved, we hope to uncover more in future work!

Results: Page 13; Lines 272-276... *"Based on these baseline measurements, our K^+ imaging suggests that relatively low $[K^+]_{in}$ may be required for presynaptic $Ca_v1.3$ channel activation in neuromast hair cells. Based on the Nernst potential, lower $[K^+]_{in}$ in active cells would lead to more depolarized membrane potentials. In active cells these potentials may be optimal for $Ca_v1.3$ channel activation and presynaptic function."*

Discussion: Page 17; Lines 360-367... *"Mechanistically, in our in vivo study, low hair-cell $[K^+]_{in}$, and subsequently more depolarized membrane potentials are important for presynaptic $Ca_v1.3$ channel activity. In support of this, in silent cells with higher $[K^+]_{in}$, despite properly localized $Ca_v1.3$ channels (Fig. 5a-b), depolarization is unable to activate these channels (Fig. 5c). Studies on these channels indicate that $Ca_v1.3$ channel open probability and open time occur optimally within a membrane voltage range from -60 to -40 mV[5, 6]. Based on this work, and the Nernst equilibrium potential, increased $[K^+]_{in}$ in silent cells may reflect hyperpolarized membrane potentials that place the cell out of this optimal activation range."*

Discussion: Page 17; Lines 368-385... *"Although low hair-cell $[K^+]_{in}$ is important for presynaptic activity, it remains unclear how $[K^+]_{in}$ heterogeneity is set up within a neuromast organ. Although we found that surrounding supporting cells function broadly to maintain low hair-cell $[K^+]_{in}$ levels, they appear unable to maintain all hair cells within each neuromast with sufficiently low $[K^+]_{in}$ to facilitate presynaptic activity. Therefore, it is possible that differences in $[K^+]_{in}$ is set up by intrinsic activity differences among hair-cells. In neurons, increased excitability has been attributed to increases in Na^+/K^+ ATPase activity resulting in more depolarized membrane potentials, or to increased resistance due to a decrease in K^+ leak channel current [7]. Both of these mechanisms increase excitability through a corresponding increase in $[K^+]_{in}$. Because K^+ rather than Na^+ is the carrier ion for the depolarizing current in hair cells[8], and because our study associated active cells with low $[K^+]_{in}$, it is unlikely that either of these mechanisms increase excitability in neuromast hair cells. Alternatively, it is possible that differences in K^+ channel properties such as BK (KCNMA1) or SK (KCNQ4) channels could regulate K^+ efflux and resting $[K^+]_{in}$ among hair cells. Consistent with this idea, in mice, SK channels have been associated with the maintenance of hair-cell resting potentials[9]. In zebrafish both BK and SK channels have been shown to be expressed in the zebrafish inner ear[10, 11], and transient knockdown of BK channels resulted in larvae with diminished hearing[10]. Overall, work on the regulation of hair-cell excitability via $[K^+]_{in}$ levels warrants further investigation."*

Discussion: Page 20; Lines 422-430... *“The presence of silent cells suggests that not all cells are needed for proper lateral-line function and that silent cells exist as backups. In support of this, previous work found that after moderate damage to hair cells in the lateral line, no quantifiable behavioral deficit was observed[12]. Therefore, anatomical redundancy and presynaptic silencing may work together to create a sensory system that functions with the minimum number of cells and connections. This system could be beneficial in several ways. First, if all cells are not needed, silencing the majority of cells could prevent unnecessary energy loss and cellular stress. Second, silent cells could rapidly backup cells lost after damage—this is especially important in sensory systems which are critical for larval zebrafish survival[13].”*

Reviewer’s concern (Main Comments 4): *“It is unclear how gap junctions can lower intracellular potassium in hair cells. Is potassium entering via mechanotransduction channels and leaving through voltage-gated channels to be sucked up by supporting cells? How does it get into supporting cells, transported? Did you investigate K levels in supporting cells or voltage of supporting cells? Are cells with lower potassium depolarized because of the shift in Nernst potential? Would depolarization make the cells more or less responsive to hair bundle deflection? This could be argued either way as input resistance is likely lower as is driving force and calcium channel inactivation likely more prevalent, all contributing to reducing sensitivity of the hair cells. Alternatively, a smaller receptor potential might be more effective if the cell’s resting potential overlaps with the calcium channel activation curve.”*

“It is unclear how gap junctions can lower intracellular potassium in hair cells. Is potassium entering via mechanotransduction channels and leaving through voltage-gated channels to be sucked up by supporting cells? How does it get into supporting cells, transported?”

Response: In our revised manuscript, based on the supporting cell and glia literature we have done our best to outline how supporting cells might take up, buffer and remove K⁺. It is true that gap junctions cannot accomplish this on their own.

Figure 9: This figure outlines a model of the molecules that have been shown to be important for glia and supporting cell K⁺ buffering and recycling. It is possible that similar molecules may be working in neuromast organs. Future work examining hair-cell activity in these mutants will help determine their exact roles.

Discussion: Page 18; Lines 403-409... *“For K⁺ buffering in supporting cells, gap junctions alone are not sufficient. Similar to glia, supporting cells may use KIR_{4.1} channels, along with the K⁺-Cl⁻ co-transporters KCC3 and KCC4, to move K⁺ in and out of the syncytia[14] (Fig. 9). In support of this, lesions in Connexin26 (Cx26 subunits form a gap junction channel), KIR_{4.1} and KCC3/4 genes have all been associated with hearing loss in humans or mice[15-17]. While the homologues of many of these genes are expressed in zebrafish hair-cell organs[18, 19], additional work is needed to determine if they function in zebrafish supporting cells to promote synaptic activity in neuromast hair cells.”*

We have proposed that K⁺ levels the hair cells during depolarization, leading to an increase in external K⁺ that is taken up by supporting cells.

Discussion: Pages 18-19; Lines 399-400... *“K⁺ buffering is important in hair-cell organs, as increases in extracellular K⁺ have been observed during stimulation of hair cells[20, 21].”*

Did you investigate K levels in supporting cells or voltage of supporting cells?

Response: Although we are currently unable to measure voltage in supporting cells, we did measure internal K⁺ in supporting cells.

Figure 8e: Using APG-2, we measured internal K⁺ levels in supporting cells. On average, it was lower than in hair cells. Importantly, block of gap junctions with FFA significantly increased internal K⁺ in supporting cells as well as hair cells. This indicates that K⁺ recycling is disrupted in supporting cells upon gap junction block.

Results: Page 15; Lines 326-328... *“We observed that after FFA application, $[K^+]_{in}$ levels were significantly elevated in hair cells (Fig. 8c-d) as well as in supporting cells (Fig. 8e). This indicates that blocking gap junctions may impair the ability of supporting cells to buffer and remove $[K^+]_{ex}$.”*

Are cells with lower potassium depolarized because of the shift in Nernst potential?

Response: In our results and discussion, we propose that cells with lower internal K⁺ are more depolarized due to a shift in the Nernst potential, see main comment #3 above.

Would depolarization make the cells more or less responsive to hair bundle deflection? This could be argued either way as input resistance is likely lower as is driving force and calcium channel inactivation likely more prevalent, all contributing to reducing sensitivity of the hair cells. Alternatively, a smaller receptor potential might be more effective if the cell's resting potential overlaps with the calcium channel activation curve.”

Response: Overall, it appears that on average, depolarization makes hair cells more responsive to hair-bundle deflection. We also agree and discuss the impact of a more depolarization on the calcium channel activation curve, see main comment #3 above.

Reviewer's concern (Other Comments 1): *“Abstract and intro: authors spend more time than necessary trying to justify the model system. This seems unnecessary and somewhat irrelevant to their story. Lateral line is not a model for mammalian cochlea and shouldn't be presented that way, it is an important system on its own.”*

Response: Good point. Based on the reviewer's suggestion, we have limited our comparisons to mammals and other models and tried to focus on what is known in zebrafish and how it might specifically impact neuromasts and lateral-line function.

Reviewer's concern (Other Comments 2): *“Please define suprathreshold mechanical stimulation, term is vague and so uninformative”*

Response: For clarity, in the revised abstract, we have swapped out suprathreshold with saturating. This mechanical stimulation is the stimulus strength that maximally stimulates the hair cell without damaging the hair bundle.

Reviewer's concern (Other Comments 3): *"Please be consistent when discussing low resting potassium that you are referring to internal K and not extracellular K"*

Response: In our revised manuscript, we have aimed for consistency with our K⁺ nomenclature, and denote intracellular K⁺ as [K_{in}], and extracellular K⁺ as [K_{ex}] to clarify this issue.

Reviewer's concern (Other Comments 4): *"Too much time spent on justifying a model system, particularly as representative of other inner ear organs as this is simply not true at most levels. Should simply discuss as mechano detector in context with what the lateral line actually does (this is really overlooked). Line 66 amicable is a strange choice of words."*

Response: In our revised manuscript, we have limited our comparisons to other inner ear organs. In the discussion, we also discuss the possible relevance of the redundant anatomy silent cells on lateral-line function, see above discussion on plasticity in main comment #3. Amicable has been removed.

Reviewer's concern (Other Comments 5): *"Line 86, what does 'complete mechanotransduction' mean?"*

Response: We were trying to convey that perhaps to be defined as mechanotransduction, both apical mechanosensation and synaptic activity are required. The way this is stated is confusing in the text, especially a non-hair cell scientist. In this revision, we changed the wording.

Results: Pages 3-4; Lines 72-74... *"Our experimental results show that when stimulated together, although all hair cells within a neuromast organ are mechanosensitive, the majority of hair cells are synaptically silent with no presynaptic Ca²⁺ influx, vesicle fusion or associated postsynaptic activity."*

Reviewer's concern (Other Comments 6): *"Line 92 What does redundant mean? How are silent cells redundant? Do they not serve to replace damaged cells, is this not an important role that might be specialized to a lateral line system where rapid replacement is necessary for survival, i.e. regeneration would be too slow."*

Response: We agree that the label "redundant" is confusing. In our revision, we have done our best to distinguish anatomical redundancy versus a reserve of silent cells, as the silent cells are more of a reserve than a redundant backup. See above discussion on plasticity under main comment #3.

Reviewer's concern (Other Comments 7): *"Line 109 Enriched typically means higher concentration per unit area, is this what the authors mean or are they saying that the surface area to volume ratio is greater in the hair bundle so the apparent level of indicator is higher?"*

Response: Our goal was to convey that because GCaMP6s is in the membrane, it can be used to detect bundle signals. In previous zebrafish work in hair cells, transgenic lines used cytosolic calcium indicators and could not detect subcellular bundle signals. Compared to cytosolic indicators, the GCaMP6s-CAAX indicator is indeed brighter in the apical bundles. The text has been condensed due to the word limit and reads as follows:

Results: Page 4; Lines 86-94... “To understand the functional contribution of all the individual hair cells in a neuromast organ, we used a transgenic zebrafish line expressing a hair-cell-specific, plasma membrane targeted Ca^{2+} indicator GCaMP6s-CAAX[22] (Fig. 1b, Supplementary Fig. 1a). In response to mechanical stimuli, this line can be used to examine mechanosensitive ion channel Ca^{2+} influx[23, 24] in apical mechanosensory bundles. In addition, it can be used to detect subsequent opening of presynaptic, voltage-gated Ca^{2+} channels ($Ca_v1.3$) at the base of hair cells[25]. Pharmacological controls confirmed that this line reliably detects hair-cell Ca^{2+} signals within these two distinct functional compartments (See methods, Supplementary fig. 2).

Methods: Page 28; Lines 602-608... “To confirm that the apical, mechanosensitive Ca^{2+} responses detected using *myo6b:GCaMP6s-CAAX* were not due to motion artifacts but activities of mechanosensitive ion channels present in mechanosensory bundles, we applied BAPTA abolish mechanosensitivity. This eliminated mechanosensitive Ca^{2+} responses at the apex as well as presynaptic Ca^{2+} responses at the base (Supplementary fig. 2a-b'). To confirm that presynaptic Ca^{2+} responses at the base of the hair cell were dependent on $Ca_v1.3$, we applied the L-type Ca^{2+} channel antagonist isradipine. Isradipine eliminated all presynaptic Ca^{2+} responses at the base of the hair cell, with no impact on apical mechanosensitive signals (Supplementary fig. 2c-d').”

Reviewer's concern (Other Comments 8): “Lines 119-122 is there any attempt to quantify the level of calcium entry per bundle as a means of assessing relative effects on receptor potential? Are the signals all comparable or do some get brighter or stay brighter longer than others?”

Response: With our current approaches, unfortunately it is not possible to get a readout of the amount of calcium entering each bundle, or a direct readout of how calcium relates to the mechanosensitive element of the receptor potential. In the future, it will be exciting to pair electrophysiological recording with bundle calcium measurements. For now, we have done our best to more closely show the features that characterize the bundle calcium signals in active and silent cells.

Figure 1b'': Now instead of just heat maps to denote bundle calcium signals, we also show traces of the apical bundle responses for all the cells within the example neuromast, showing the bundle responses of active and silent cells. Some responses are larger than others. All responses stay elevated during the stimulation window. It appears that larger responses may take longer to reach baseline levels, but it is unclear if this is due to the kinetics of the indicator or due to properties of individual hair cells.

Reviewer's concern (Other Comments 9): “Is there any difference in baseline levels of calcium either in the stereocilia or basolaterally that might support an argument for cells being more depolarized or having more active MET?”

Response: Previously we examined the baseline GCaMP6s intensity basolaterally and did not find a difference between active and silent cells. These data are still in the results (Page 13, lines 270-272) and in Supplementary Fig. 7h'.

As suggested, we also examine the baseline bundle GCaMP6s intensity more closely, and found that active cells had slightly higher, yet significant increases in baseline intensity compared to

silent cells. It equally possible that increased GCaMP6s intensity could be a result of more depolarized potentials or on average more active MET (also see other Comment 10 below).

Supplementary Fig. 7h: The resting bundle signals are plotted in active and silent cells.

Results: Pages 12-13; Lines 268-271... *"In contrast, using GCaMP6s, we were not able to detect a significant difference in the baseline Ca^{2+} levels at the presynapse between the two hair-cell populations (Supplementary fig. 7h'), although Ca^{2+} levels were slightly elevated in the hair bundles of active cells (Supplementary fig. 7h)."*

Reviewer's concern (Other Comments 10): *"Is there any correlation between the level of calcium signal apically to that basolaterally? Is there a threshold level of MET that needs to happen to trigger a basolateral effect? Line 152 suggests not but there is no quantitation presented or description of how this was measured. This needs to be included."*

Response: As suggested, we added additional quantification to compare the magnitude of apical and basolateral signals. The overall correlation of $R^2 = 0.26$ is significant and this information has been added to the results. In addition, we quantified the magnitude of the bundle calcium signals in active and silent cells more closely. In Figure 1c-c'.

Figure 1c-c': We did find that on average, active cells had larger bundle signals compared to silent cells. This could indicate that more apical calcium or MET dictates whether there is a presynaptic calcium signal. This is not always the case. The magnitude of the bundle response does not always predict whether there is a synaptic calcium response, see example Figure 1b'', cell X and the spread of bundles responses in Figure c' among active cells.

Results: Page 6; Lines 117-128... *"Amongst cells with (active) and without (silent) presynaptic Ca^{2+} signals, we observed a slight, yet significant correlation between the magnitude of the mechanosensory Ca^{2+} response versus the presynaptic Ca^{2+} response (Fig. 1b2'', b3'', and c: $R^2 = 0.26$, $p = 0.001$, $n = 94$ cells). We examined the relationship between mechanosensitive and presynaptic Ca^{2+} more closely by using a presynaptic GCaMP6s $\Delta F/F$ threshold of 20 %, the approximate off-ribbon response magnitude, to separate active and silent cells (Fig. 1c, gray bar). Using this metric, we observed that mechanosensory Ca^{2+} responses were on average significantly larger in active cells (Fig. 1c'). Although the magnitude of the mechanosensitive response was on average larger in active cells, the presence or magnitude of the mechanosensitive response did not always dictate whether there was a corresponding presynaptic Ca^{2+} response (See example Fig. 1b2'' versus b3'' and spread in c'). Overall these results indicate that although all hair cells are able to detect stimuli, only a subset of hair cells have consequent presynaptic Ca^{2+} signals."*

Figure 2e-e''': To further support our argument that MET does not always dictate presynaptic activity, this data has been moved from the supplement and placed in Figure 2. Here we examined synaptic response for a variety of stimuli, including a long 6s mechanical stimulus. Even during this strong stimulus, in silent cells, we do not observe basolateral signals above baseline. This provides further indication that it is not as simple as meeting a threshold level of MET. This is also shown using SypHy in Figure 2c' using an intense 10um bundle deflection.

It is also possible that larger bundle calcium signals are a further reflection of lower internal K⁺ in active cells.

Reviewer's concern (Other Comments 11): *“Do all hair cells have calcium channels clustered at synapses. This needs to be definitively answered for any interpretation of mechanism or function.”*

Response: Yes. See above response to main comment #2.

Reviewer's concern (Other Comments 12): *“Line 174 How is a saturating mechanical stimulation defined?”*

Response: In this revision, we clarify that a stimulus is saturated when additional deflection is unable to increase the magnitude of the apical and basal calcium signals, without damaging the hair bundle.

Methods: Page 27; Lines 566-575... *“Apical hair-bundle deflection was monitored by measuring the displacement distance of tips of the bundles, the kinocilia. For a 2-s deflection (step or 2-s 5 Hz), we found that when the kinocilial tips were deflected more than 5 μm, the magnitude of the apical and synaptic GCaMP6s Ca²⁺ signals were saturated. Within 1-5 μm kinocilial deflection distances, Ca²⁺ signals increased with increasing deflection distance and the Ca²⁺ signals were not saturated. Intense bundle deflections, that moved kinocilial tips more than 10 μm commonly induced movement artifacts, and were damaging to hair bundles. Consistent with damage, after intense 10 μm bundle deflections, subsequent Ca²⁺ responses were dramatically reduced. A 5 μm deflection distance was used in the majority of our experiments to achieve maximal Ca²⁺ signals.”*

Reviewer's concern (Other Comments 13): *“What is the innervation pattern of hair cells with respect to functional synapses? Are all afferent fibers innervated by some percentage of functional synapses or are there afferent fibers that are not mechanically sensitive? This is a critical question as it relates to both physiological relevance as well as potential mechanisms.”*

Response: Yes, the innervation pattern is extremely relevant, and we have added additional information on this. The current literature suggests that each afferent innervates nearly all hair cells of one polarity and there is no evidence that there are fibers that are not mechanically sensitive. See the response above to main comment #1.

Reviewer's concern (Other Comments 14): *“Line 311, what does mainly detected mean? Should this be quantified. Were there synapses that did not have major calcium responses that still showed vesicle fusion?”*

Response: We agree that “mainly detected” is confusing. The red, RGECO1 indicator is cytosolic, while the green, GCaMP6s indicator that we used for the majority of our work is membrane localized. There are several limitations to using RGECO1. First, RGECO1 is cytosolic so it not as accurate as GCaMP6s at detecting presynaptic calcium signals. Also, when using RGECO1, the ribbon plane is not known. Lastly, RGECO1 responses can be influenced by the magnitude of the MET signals or calcium stores. These features makes it difficult to precisely know what absolute RGECO1 response magnitude represents an active cell. We have done 2-color GCaMP6s/RGECO1 imaging in hair cells and found that RGECO1 calcium signals are overall a good indicator of

whether there is a GCaMP6s presynaptic response (data not shown). To avoid confusion, we have removed “mainly detected” and stuck with our quantification which we feel is robust.

Results: Page 9; Lines 191-202... “Therefore to directly test whether presynaptic Ca^{2+} influx, vesicle fusion and afferent activity were associated with the same subset of hair cells within a given neuromast organ, we performed two-color functional imaging. For these experiments, we used a transgenic line that expresses the red-shifted cytosolic Ca^{2+} indicator RGECO1 in hair cells,[26] in combination with either the SyHy (green) in hair cells, or GCaMP6s (green) in afferent neurons (Fig. 3c1 and Supplementary fig. 1f, g, 5a1). We found that RGECO1 Ca^{2+} responses in hair cells associated with afferent activity were larger (70%) than the responses in cells without afferent activity (Fig. 3d, example 3c1-c4”). Similarly, RGECO1 and Ca^{2+} responses were significantly larger (77%) in hair cells with vesicle fusion (Supplementary fig. 5b, example Supplementary fig. 5a1-a4”). Overall, these results confirm that active hair cells with strong presynaptic Ca^{2+} responses are indeed the same subset of cells with vesicle fusion and associated postsynaptic activity.”

Reviewer’s concern (Other Comments 15): “Why not simply depolarize the entire neuromast with potassium to see if all hair cells could be activated.”

Response: Based on this suggestion, we have used high K^+ to stimulate the entire neuromast and only active cells showed presynaptic calcium signals, see above discussion in main comment #3.

Reviewer’s concern (Other Comments 16): “The ablation experiment is interesting and valuable as it perhaps points to the function of having nonresponsive hair cells that can rapidly assume function. Were there any behavioral tests done to show that in fact there was compensation at the systems level? Were there any behavioural tests to show that there was a functional deficit when a hair cell was ablated?”

Response: This is an interesting question, and currently something that we have not found a good way to test directly. But, based on lateral-line literature, a previous study indicated that moderate damage to the hair cells in the zebrafish lateral line did not impact lateral-line function. This suggests that perhaps not all hair cells are required for function and that compensation may occur after hair-cell damage. This is also discussed above in main comment #3.

Reviewer’s concern (Other Comments 17): “The ten-minute time frame is interesting but it is unclear that it precludes a change in protein expression or distribution, in particular as it relates to the calcium channels. Channels may be expressed but not inserted and a shift can occur in this time frame.

Response: It is true that other mechanisms could occur within the 10 minute time frame. Our current discussion does not imply what can or cannot occur within this window.

Reviewer’s concern (Other Comments 18): “The efferent experiments were done well and relatively thoroughly but it is unclear that this was not a straw man type of set of experiments as the means by which the efferent system could produce this result is unclear. Are innervation patterns such that individual fibers would be selectively diminishing activity in a population of hair cells? Is there any

correlation between efferent patterns and patterns of hair cell functionality? It is somewhat unclear as to why this direction was chosen for investigation.”

Response: The efferents are an obvious and tempting candidate due to their proximity to the hair cells, especially the cholinergic efferents that directly contact the hair cells at their base. In the lateral line, currently it is not clear what efferent do or what their exact innervation patterns are, so it is hard to predict their role. Controlling hair-cell activity patterns within a neuromast or regulating internal K⁺ levels could have been their role, but our data suggests that this is not the case. Because this is negative data, we have tried to minimize our efferent results.

Reviewer’s concern (Other Comments 19): *“Does the loss of efferent/afferent prevent the recovery of functional hair cells after hair cell ablation? That is could either of these pathways be involved in detection or replacement of lost hair cells as compared to the establishment of the hair cell functional pattern?”*

Response: The efferents are also poised to detect or replace hair cells after laser damage. To test this, we redid the laser damage experiments in the *neurog1a* mutants and found that after damage, silent cells can could still become active. This indicates that innervation is not required for the recovery.

Results: Pages 13-14; Lines 288-291...*“We also used neurog1a mutants to test whether innervation was required for the conversion of silent cells into active cell after laser damage. Similar to wild-type animals, after damage to an active cell, in neurog1a mutants we observed that silent cells could become active (n = 5/7 trials).”*

Reviewer’s concern (Other Comments 20): *“A single TEM image is not really appropriate for defining an expression pattern within the neuromast. The distribution and expression level would be more appropriate. Also, how close do membranes need to be to guarantee that it represents a gap junction? Shouldn’t they be directly abutting each other and not simply closer together?”*

A single TEM image is not really appropriate for defining an expression pattern within the neuromast.

Response: We now show 5 independent example of gap junctions between supporting cells using TEM. Overall, our TEM examination was very thorough, and we examined all membranes along hair cells and supporting cells.

Figure 8a’, b and Supplementary Fig. 7a, b, c, & d: Cumulatively, 5 examples of gap junctions between supporting cells.

The distribution and expression level would be more appropriate.

Response: As soon as we can identify the connexin gene(s) in neuromast hair cells we plan to make antibodies to examine the distribution and levels of gap junction channels. Or tag them. This will be extremely useful, but it is beyond what we can currently examine.

Also, how close do membranes need to be to guarantee that it represents a gap junction? Shouldn’t they be directly abutting each other and not simply closer together?”

Response: We have now clearly defined how close together membranes must be to represent a gap junction.

Methods: Pages 25-26; Lines 540-548... *“Gap junctions were identified by established criteria for standard TEM, two cell membranes aligned parallel and relatively straight, with an intercellular gap of 2-3 nm (~5 nm including the outer stained leaflets of the two cell membranes)[27-30]. The appearance of published high magnifications of the gap junction substructure is variable, and dependent on fixation, embedding, sectioning, staining, and imaging[27, 28].... The gap junctions that we identified in the neuromasts appear to be identical in fine substructure to those found between supporting cells in the vestibular or auditory epithelium of vertebrates[29-32].*

Reviewer’s concern (Other Comments 21): *“What is the Kd of the potassium sensor? Was this calibrated with fluorescent output to get an idea of what the concentration difference is for better estimates of Nernst potential changes?”*

Response: The Kd of the potassium sensor (APG-2) is 18 mM, and this has been added to the methods. The concentration values a dye can reliably measure are usually between 0.1 Kd and 10 Kd, so 1.8 to 180 mM K⁺. Unfortunately, it was not possible to calibrate this dye to get a more accurate sense of the actual internal K⁺ concentration in hair cells. This does make it challenging to estimate the relative Nernst potential differences.

Reviewer’s concern (Other Comments 22): *“Line 446 need to be clear as to intra or extra cellular potassium levels.”*

Response: We are now much clearer with regards to intra- and extra-cellular K⁺ levels in the text.

Reviewer’s concern (Other Comments 23): *“The discussion is weak as it includes a lot of vagueness about potential mechanisms and about potential function. It also attempts to discuss other hair cell organs which may or may not be relevant but it excludes discussing potential functional relevance to the lateral line organ. There are no clear feasible mechanisms discussed nor tested. The functional relevance is speculative and not well founded.”*

Response: We have made significant modifications to the discussion in this revision and it is much more concise. In addition, we have done our best to keep our focus on what could be happening in neuromast organs and why.

Response to Reviewer 2

Reviewer’s concern (Main Comments 1): *“Selective inhibition of gap junctions is difficult to achieve pharmacologically. FFA and heptanol have many other effects (particularly on chloride channels). Genetic manipulation of gap junctions in supporting cells would provide much stronger evidence for their involvement.”*

Response: Gap junction inhibition is indeed difficult to achieve. We did use both FFA and heptanol which due to their diverse structures are thought to work via slightly different

mechanisms. To achieve maximum specificity with each of these drugs, we kept our doses as low as possible, as higher concentrations lead to further loss of specificity.

In addition, in the case of FFA, the possibility of off target block of chloride channels is a good point. This is a real concern for our study, because chloride channels have been shown to be important for supporting-cell function in mammals during development. To rule out a rule out off-target chloride channels block, we tested 4 chloride channel blockers NPPB, T16A(inh)-A01, DIDS and benzbromarone. DIDS and benzbromarone completely block mechanosensitive calcium responses, and therefore we were not able to use them to assess presynaptic calcium responses. NPPB and T16A(inh)-A01 both reduced presynaptic calcium responses, but not to the same degree as FFA or heptanol. Overall this result is difficult to interpret, as Cl⁻ channels could also be important for presynaptic function or they could be part of the FFA block. We have added information on Cl⁻ channel block to our results. Although this does not strengthen our argument regarding gap junctions, it does lend further support to the role of supporting cells in facilitating presynaptic calcium responses.

Supplementary Fig. 8i: Shows that NPPB does not alter presynaptic calcium responses.

Results: Pages 15; Lines 318-323... *“The gap junction antagonist FFA has been shown to have other targets including Cl⁻ channels, which are also important for supporting-cell function in mammals[33]. Therefore, we applied the Cl⁻ channel blocker T16A(inh)-A01, and observed a 30 % reduction in presynaptic Ca²⁺ signals (Supplementary fig. 7i). Based on these results, we conclude that supporting cells may use both gap junctions as well as Cl⁻ channels to facilitate presynaptic activity in hair cells.”*

We also agree that creating mutants in connexins genes in the zebrafish is the best way to back up our pharmacological results. Unfortunately, while there are 20 Cxs in mammals, there are about 40 in zebrafish. In addition, the exact homologue for Cx26 (one of the most important Cx's in the mammalian inner ear) is not clear. Currently, we have made CRISPR mutants for the 6 of the most likely Cx candidates. Thus far all single mutants are normal with regards to hair-cell function, and we are currently testing double mutant combinations to address this question/issue further. Unfortunately, this data will have to be a part of a future study.

Reviewer's concern (Main Comments 2): *“Missing from the study is any assessment of hair cell membrane potential. Knowledge about these changes will be crucial to provide further evidence for the K⁺ hypothesis. For example, it would be nice to know if latent hair cells experience the same amount of depolarization in response to hair bundle deflection. Is it possible that these cells have lower membrane resistance? If so, one might assume that inhibiting K⁺ channels would promote recruitment of latent cells. If genetically encoded voltage sensors work in this system it would provide a nice complement to the calcium imaging, as it would help define how depolarization-excitation coupling is rapidly altered. In my mind, it would be sufficient to just speculate about some potential mechanisms in the discussion.”*

Response: We agree 100% – addressing or acknowledging relative differences resting membrane potential and understanding if hair-cell depolarization activates Cav1.3 channels was a critical element missing from our study. This was also a major concern for Reviewer #1.

Based on this input we have done our best to assay hair-cell depolarization, detect voltage changes, and speculate about resting membrane potential differences in the discussion.

For more info, please see the comments from Reviewer #1, main concern #3.

We are also still working hard to find a way to use voltage indicators, such as Bongwoori to get a readout of resting membrane potential. Thus far we have not found a way to do this in combination with other functional imaging to separate active and silent cells.

Reviewer's concern (Main Comments 3): *"The authors suggest that supporting cell gap junctions somehow create local K⁺ gradients that differentially alter the excitability of hair cells. This is an unconventional view of the role of gap junctions, as gap junctions are typically thought to normalize ion gradients. One would expect that they would help to buffer K⁺, dissipating transients generated by active cells through the glial syncytium, rather than produce persistent local increases. It would be helpful if the authors were to provide a model to explain how they think supporting cells influence hair cell excitation by this mechanism."*

Response: In our previous manuscript, we agree that we were too vague in our conclusions regarding the role of gap junctions, K⁺, and hair-cell excitability. In our revised manuscript, we have attempted to clarify our conclusions. We now suggest that perhaps gap junctions are required more broadly to buffer K⁺ in order to lower K⁺ levels in all hair cells, rather than creating local K⁺ gradients. Although gap junctions could create local gradients, we have little data to support it, and in all part of the manuscript, we now focus on a more general role for gap junction channels.

Examples:

Introduction: Page 4, lines 77-80... *"Our pharmacological results indicate that networks of glia-like supporting cells that surround neuromast hair cells may employ gap junctions to buffer and remove [K⁺]_{ex} maintain low [K⁺]_{in} in hair cells."*

Results: Page 15, lines 327-330... *"This indicates that blocking gap junctions may impair the ability of supporting cells to buffer and remove [K⁺]_{ex}. Removal of [K⁺]_{ex} and maintenance of low [K⁺]_{in} levels in hair cells is critical to facilitate synaptic activity."*

Discussion: Page 17, lines 368-373... *"Although low hair-cell [K⁺]_{in} is important for presynaptic activity, it remains unclear how [K⁺]_{in} heterogeneity is set up within a neuromast organ. Although we found that surrounding supporting cells function broadly to maintain low hair-cell [K⁺]_{in} levels, they appear unable to maintain all hair cells within each neuromast with sufficiently low [K⁺]_{in} to facilitate presynaptic activity. Therefore, it is possible that differences in [K⁺]_{in} are established by intrinsic activity differences among hair-cells."*

Discussion: Page 18, lines 394-396... *"Although it is unclear if supporting cells play a direct role in unsilencing cells after laser damage, our in vivo work indicates that under steady-state conditions, supporting-cell syncytia are essential to buffer [K⁺]_{ex} maintain low hair-cell [K⁺]_{in}."*

Reviewer's concern (Main Comments 4): *"In the experiments that have been performed, the functional consequence of the different intracellular K⁺ levels has not been established (it is not clear how much K⁺ varies between these cells or how inhibiting supporting cell gap junctions would lead to changes in intracellular K⁺ in hair cells). In addition, the studies in which extracellular K⁺ was changed are complicated by the fact that this manipulation will also induce depolarization and perhaps inactivation of calcium channels."*

Response: It is true, we are unable to establish how much K⁺ varies between cells. This is definitely a limitation with the dye. In the future, it is our hope that better dyes will allow us to calibrate and determine the relative K⁺ differences among hair cells.

Figure 9. We have tried to illustrate in this figure how gap junctions, along with other accessory molecules could function to overall lower internal K⁺ in hair cells and facilitate presynaptic activity.

We originally increased extracellular K⁺ levels to verify the K⁺ dye APG-2 was functional. In the context of this work, this data complicates our results, and is difficult to interpret because, as suggested by reviewer #2, elevated K⁺ could block of calcium channels or create some other form of depolarization block. For this reason, we have removed this data from the study.

Reviewer's concern (Other Comments 1): *"In Fig. 1b4 – the line colors do not match the legend"*

Response: Thank you for pointing this out, the line color now matches the legend!

Reviewer's concern (Other Comments 2): *"The pseudocoloring in Fig. 1 should be standardized for the two experiments to facilitate comparisons (and start from 0)."*

Response: When using $\Delta F/F_0$ it is challenging to start color maps at 0 because by starting everything at zero noise becomes a real issue. This makes it difficult to visualize the signals above noise. This issue does not happen when using ΔF to generate color maps - which many people use - because it illustrates relative changes rather than changes from 0. Overall, we prefer to use $\Delta F/F_0$ so our color maps match our plots. For Figure 1 we have tried to be as consistent as possible. We used the same starting value for the apex and base, and have included response plots so the response magnitudes are easier to compare.

In an attempt to be as consistent as possible throughout our figures, we determined the noise floor (bottom of the color map) for each indicator and kept it consistent throughout the figures. An exception is Figure 4, as the data was acquired on separate wide field system. When possible we also tried to keep the maximum value on our heat maps as consistent as possible.

Reviewer's concern (Other Comments 3): *"Reword: While we observed that fewer hair cells with vesicle fusion during a 50 Hz stimulus, for all other stimuli tested, a similar percentage of hair cells per neuromast had vesicle fusion (Figure 2c, one-way ANOVA, $df = 30$)."*

Response: Good suggestion.

Results: Pages 7-8, lines 158-161... *"While we observed fewer hair cells with vesicle fusion and presynaptic Ca²⁺ responses during a 50 Hz stimulus (Fig. 2c, e'), for all other stimuli tested, a similar percentage of hair cells per neuromast had vesicle fusion (Fig. 2c), and the same subset of hair cells had robust presynaptic Ca²⁺ signals (Fig. 2e-e''''")."*

References:

1. Kindt, K.S., G. Finch, and T. Nicolson, *Kinocilia mediate mechanosensitivity in developing zebrafish hair cells*. *Dev Cell*, 2012. **23**(2): p. 329-41.
2. Jiang, T., K. Kindt, and D.K. Wu, *Transcription factor Emx2 controls stereociliary bundle orientation of sensory hair cells*. *Elife*, 2017. **6**.
3. Sheets, L., K.S. Kindt, and T. Nicolson, *Presynaptic CaV1.3 channels regulate synaptic ribbon size and are required for synaptic maintenance in sensory hair cells*. *J Neurosci*, 2012. **32**(48): p. 17273-86.
4. Pujol-Marti, J., et al., *Converging axons collectively initiate and maintain synaptic selectivity in a constantly remodeling sensory organ*. *Curr Biol*, 2014. **24**(24): p. 2968-74.
5. Koschak, A., et al., *alpha 1D (Cav1.3) subunits can form l-type Ca²⁺ channels activating at negative voltages*. *J Biol Chem*, 2001. **276**(25): p. 22100-6.
6. Xu, W. and D. Lipscombe, *Neuronal Ca(V)1.3alpha(1) L-type channels activate at relatively hyperpolarized membrane potentials and are incompletely inhibited by dihydropyridines*. *J Neurosci*, 2001. **21**(16): p. 5944-51.
7. Berndt, N. and H.G. Holzhammer, *The high energy demand of neuronal cells caused by passive leak currents is not a waste of energy*. *Cell Biochem Biophys*, 2013. **67**(2): p. 527-35.
8. Art, J.J. and R. Fettiplace, *Variation of membrane properties in hair cells isolated from the turtle cochlea*. *J Physiol*, 1987. **385**: p. 207-42.
9. Kharkovets, T., et al., *Mice with altered KCNQ4 K⁺ channels implicate sensory outer hair cells in human progressive deafness*. *EMBO J*, 2006. **25**(3): p. 642-52.
10. Rohmann, K.N., et al., *Manipulation of BK channel expression is sufficient to alter auditory hair cell thresholds in larval zebrafish*. *J Exp Biol*, 2014. **217**(Pt 14): p. 2531-9.
11. Wu, C., et al., *Kcnq1-5 (Kv7.1-5) potassium channel expression in the adult zebrafish*. *BMC Physiol*, 2014. **14**: p. 1.
12. Niihori, M., et al., *Zebrafish swimming behavior as a biomarker for ototoxicity-induced hair cell damage: a high-throughput drug development platform targeting hearing loss*. *Transl Res*, 2015. **166**(5): p. 440-50.
13. Nicolson, T., et al., *Genetic analysis of vertebrate sensory hair cell mechanosensation: the zebrafish circler mutants*. *Neuron*, 1998. **20**(2): p. 271-83.
14. Wan, G., G. Corfas, and J.S. Stone, *Inner ear supporting cells: Rethinking the silent majority*. *Seminars in Cell & Developmental Biology*, 2013. **24**(5): p. 448-459.
15. Kelsell, D.P., et al., *Connexin 26 mutations in hereditary non-syndromic sensorineural deafness*. *Nature*, 1997. **387**(6628): p. 80-3.
16. Boettger, M.K., et al., *Calcium-activated potassium channel SK1- and IK1-like immunoreactivity in injured human sensory neurones and its regulation by neurotrophic factors*. *Brain*, 2002. **125**(Pt 2): p. 252-63.
17. Everett, L.A., et al., *Pendred syndrome is caused by mutations in a putative sulphate transporter gene (PDS)*. *Nat Genet*, 1997. **17**(4): p. 411-22.
18. Mahmood, F., et al., *A zebrafish model of CLN2 disease is deficient in tripeptidyl peptidase 1 and displays progressive neurodegeneration accompanied by a reduction in proliferation*. *Brain*, 2013. **136**(Pt 5): p. 1488-507.
19. Chang-Chien, J., et al., *The connexin 30.3 of zebrafish homologue of human connexin 26 may play similar role in the inner ear*. *Hear Res*, 2014. **313**: p. 55-66.
20. Johnstone, B.M., et al., *Stimulus-related potassium changes in the organ of Corti of guinea-pig*. *The Journal of Physiology*, 1989. **408**(1): p. 77-92.
21. Valli, P., G. Zucca, and L. Botta, *Perilymphatic potassium changes and potassium homeostasis in isolated semicircular canals of the frog*. *The Journal of Physiology*, 1990. **430**(1): p. 585-594.
22. Chen, T.W., et al., *Ultrasensitive fluorescent proteins for imaging neuronal activity*. *Nature*, 2013. **499**(7458): p. 295-300.
23. Fettiplace, R., *Defining features of the hair cell mechano-electrical transducer channel*. *Pflugers Archiv-European Journal of Physiology*, 2009. **458**(6): p. 1115-1123.
24. Corey, D.P. and A.J. Hudspeth, *Ionic basis of the receptor potential in a vertebrate hair cell*. *Nature*, 1979. **281**(5733): p. 675-7.

25. Moser, T. and D. Beutner, *Kinetics of exocytosis and endocytosis at the cochlear inner hair cell afferent synapse of the mouse*. Proc Natl Acad Sci U S A, 2000. **97**(2): p. 883-8.
 26. Sheets, L., et al., *Enlargement of ribbons in zebrafish hair cells increases calcium currents, but disrupts afferent spontaneous activity and timing of stimulus onset*. J Neurosci, 2017.
 27. Revel, J.P. and M.J. Karnovsky, *Hexagonal array of subunits in intercellular junctions of the mouse heart and liver*. J Cell Biol, 1967. **33**(3): p. C7-C12.
 28. Brightman, M.W. and T.S. Reese, *Junctions between intimately apposed cell membranes in the vertebrate brain*. J Cell Biol, 1969. **40**(3): p. 648-77.
 29. Nadol, J.B., Jr., et al., *Tight and gap junctions in a vertebrate inner ear*. Am J Anat, 1976. **147**(3): p. 281-301.
 30. Hama, K. and K. Saito, *Gap junctions between the supporting cells in some acoustico-vestibular receptors*. J Neurocytol, 1977. **6**(1): p. 1-12.
 31. Mulroy, M.J., et al., *Gap junctional connections between hair cells, supporting cells and nerves in a vestibular organ*. Hear Res, 1993. **71**(1-2): p. 98-105.
 32. Kikuchi, T., et al., *Gap junction systems in the rat vestibular labyrinth: immunohistochemical and ultrastructural analysis*. Acta Otolaryngol, 1994. **114**(5): p. 520-8.
 33. Wang, H.C., et al., *Spontaneous Activity of Cochlear Hair Cells Triggered by Fluid Secretion Mechanism in Adjacent Support Cells*. Cell, 2015. **163**(6): p. 1348-59.
1. Kindt, K.S., G. Finch, and T. Nicolson, *Kinocilia mediate mechanosensitivity in developing zebrafish hair cells*. Dev Cell, 2012. **23**(2): p. 329-41.
 2. Jiang, T., K. Kindt, and D.K. Wu, *Transcription factor Emx2 controls stereociliary bundle orientation of sensory hair cells*. Elife, 2017. **6**.
 3. Sheets, L., K.S. Kindt, and T. Nicolson, *Presynaptic Cav1.3 channels regulate synaptic ribbon size and are required for synaptic maintenance in sensory hair cells*. J Neurosci, 2012. **32**(48): p. 17273-86.
 4. Pujol-Marti, J., et al., *Converging axons collectively initiate and maintain synaptic selectivity in a constantly remodeling sensory organ*. Curr Biol, 2014. **24**(24): p. 2968-74.
 5. Koschak, A., et al., *alpha 1D (Cav1.3) subunits can form I-type Ca²⁺ channels activating at negative voltages*. J Biol Chem, 2001. **276**(25): p. 22100-6.
 6. Xu, W. and D. Lipscombe, *Neuronal Ca(V)1.3alpha(1) L-type channels activate at relatively hyperpolarized membrane potentials and are incompletely inhibited by dihydropyridines*. J Neurosci, 2001. **21**(16): p. 5944-51.
 7. Berndt, N. and H.G. Holzhutter, *The high energy demand of neuronal cells caused by passive leak currents is not a waste of energy*. Cell Biochem Biophys, 2013. **67**(2): p. 527-35.
 8. Art, J.J. and R. Fettiplace, *Variation of membrane properties in hair cells isolated from the turtle cochlea*. J Physiol, 1987. **385**: p. 207-42.
 9. Kharkovets, T., et al., *Mice with altered KCNQ4 K⁺ channels implicate sensory outer hair cells in human progressive deafness*. EMBO J, 2006. **25**(3): p. 642-52.
 10. Rohmann, K.N., et al., *Manipulation of BK channel expression is sufficient to alter auditory hair cell thresholds in larval zebrafish*. J Exp Biol, 2014. **217**(Pt 14): p. 2531-9.
 11. Wu, C., et al., *Kcnq1-5 (Kv7.1-5) potassium channel expression in the adult zebrafish*. BMC Physiol, 2014. **14**: p. 1.
 12. Niihori, M., et al., *Zebrafish swimming behavior as a biomarker for ototoxicity-induced hair cell damage: a high-throughput drug development platform targeting hearing loss*. Transl Res, 2015. **166**(5): p. 440-50.
 13. Nicolson, T., et al., *Genetic analysis of vertebrate sensory hair cell mechanosensation: the zebrafish circler mutants*. Neuron, 1998. **20**(2): p. 271-83.
 14. Wan, G., G. Corfas, and J.S. Stone, *Inner ear supporting cells: Rethinking the silent majority*. Seminars in Cell & Developmental Biology, 2013. **24**(5): p. 448-459.
 15. Kelsell, D.P., et al., *Connexin 26 mutations in hereditary non-syndromic sensorineural deafness*. Nature, 1997. **387**(6628): p. 80-3.
 16. Boettger, M.K., et al., *Calcium-activated potassium channel SK1- and IK1-like immunoreactivity in injured human sensory neurones and its regulation by neurotrophic factors*. Brain, 2002. **125**(Pt 2): p. 252-63.

17. Everett, L.A., et al., *Pendred syndrome is caused by mutations in a putative sulphate transporter gene (PDS)*. Nat Genet, 1997. **17**(4): p. 411-22.
18. Mahmood, F., et al., *A zebrafish model of CLN2 disease is deficient in tripeptidyl peptidase 1 and displays progressive neurodegeneration accompanied by a reduction in proliferation*. Brain, 2013. **136**(Pt 5): p. 1488-507.
19. Chang-Chien, J., et al., *The connexin 30.3 of zebrafish homologue of human connexin 26 may play similar role in the inner ear*. Hear Res, 2014. **313**: p. 55-66.
20. Johnstone, B.M., et al., *Stimulus-related potassium changes in the organ of Corti of guinea-pig*. The Journal of Physiology, 1989. **408**(1): p. 77-92.
21. Valli, P., G. Zucca, and L. Botta, *Perilymphatic potassium changes and potassium homeostasis in isolated semicircular canals of the frog*. The Journal of Physiology, 1990. **430**(1): p. 585-594.
22. Chen, T.W., et al., *Ultrasensitive fluorescent proteins for imaging neuronal activity*. Nature, 2013. **499**(7458): p. 295-300.
23. Fettiplace, R., *Defining features of the hair cell mechano-electrical transducer channel*. Pflugers Archiv-European Journal of Physiology, 2009. **458**(6): p. 1115-1123.
24. Corey, D.P. and A.J. Hudspeth, *Ionic basis of the receptor potential in a vertebrate hair cell*. Nature, 1979. **281**(5733): p. 675-7.
25. Moser, T. and D. Beutner, *Kinetics of exocytosis and endocytosis at the cochlear inner hair cell afferent synapse of the mouse*. Proc Natl Acad Sci U S A, 2000. **97**(2): p. 883-8.
26. Sheets, L., et al., *Enlargement of ribbons in zebrafish hair cells increases calcium currents, but disrupts afferent spontaneous activity and timing of stimulus onset*. J Neurosci, 2017.
27. Revel, J.P. and M.J. Karnovsky, *Hexagonal array of subunits in intercellular junctions of the mouse heart and liver*. J Cell Biol, 1967. **33**(3): p. C7-C12.
28. Brightman, M.W. and T.S. Reese, *Junctions between intimately apposed cell membranes in the vertebrate brain*. J Cell Biol, 1969. **40**(3): p. 648-77.
29. Nadol, J.B., Jr., et al., *Tight and gap junctions in a vertebrate inner ear*. Am J Anat, 1976. **147**(3): p. 281-301.
30. Hama, K. and K. Saito, *Gap junctions between the supporting cells in some acoustico-vestibular receptors*. J Neurocytol, 1977. **6**(1): p. 1-12.
31. Mulroy, M.J., et al., *Gap junctional connections between hair cells, supporting cells and nerves in a vestibular organ*. Hear Res, 1993. **71**(1-2): p. 98-105.
32. Kikuchi, T., et al., *Gap junction systems in the rat vestibular labyrinth: immunohistochemical and ultrastructural analysis*. Acta Otolaryngol, 1994. **114**(5): p. 520-8.
33. Wang, H.C., et al., *Spontaneous Activity of Cochlear Hair Cells Triggered by Fluid Secretion Mechanism in Adjacent Support Cells*. Cell, 2015. **163**(6): p. 1348-59.

Reviewers' comments:

Reviewer #1 (Remarks to the Author):

I commend the authors for a careful and thorough response to the previous comments on this manuscript. This work represents a significant body of new data that addresses an interesting and novel finding in the neuromast organ suggesting that there are a significant number of mechanically-sensitive hair cells that are synaptically inactive until damage to the neuromast, at which time these cells are recruited into functionality. Data presented that this happens is strong and supports this basic conclusion. However, the arguments about how this happens and what the causal players are in this mechanism remain somewhat confused. The majority of previous comments were addressed directly and clearly and the manuscript is much improved because of it; the remaining issues are as follows.

1- The authors argue that nonactive hair cells may be hyperpolarized so that calcium channels are not active and that K depolarization will not depolarize the cells far enough to activate these channels. Although neither the voltage dye employed nor the K sensor used can quantitatively assess the absolute membrane potentials or K levels, there does seem to be enough data presented to negate this possibility. For example, the 30% difference in K levels measured intracellularly would be predicted to have a 5-10mV effect on resting potential. The high potassium used was perfusing a 1M KCl solution. Aside from the large osmotic effect expected to accompany this perfusion, a change at the hair cell of up to 50 mM would depolarize the hair cell to around -20 mV (ostensibly the maximal calcium current activation potential in hair cells). As access of compounds to the hair cells does not seem to be an issue a 50 mM change representing a 20x dilution of the pipette solution seems pretty conservative. This extremely large change in extracellular K would maximally depolarize the hair cell irregardless of the likely difference in intracellular potassium. Thus hyperpolarization, or input resistance of the cell cannot be responsible for the differences observed. Given that the calcium channels are identified immunocytochemically, the only two real choices appear to be that the channels are not inserted into the plasma membrane and thus remain insensitive to the electric field changes or that the channels are inserted into the membrane but are biochemically inactivated. Examples of both exist in the literature and can work on the described time frame to switch a cell from synaptically inactive to active.

2- The gap junctional blockers and Cl channel blocker data remain difficult to reconcile with the fundamental finding of synaptically active and inactive cells. Does the use of these blockers make all cells synaptically active? If there is a causal link between gap-junctional clearance of potassium or Cl channel activity and these two populations of cells than you should see it with the experiments that were performed. It is not clear from the description of the results that this happens.

Minor

1- The authors argue that the hair cell recruitment from inactive to active is 'stochastic.' This word has very specific meanings and the data provided does not justify the use of this word. It is unclear how the recruitment can be random, specifically it would be very useful to know if afferent fiber innervation patterns are dictating which cells are activated as activating a hair cell randomly will not necessarily be valuable. If a hair cell whose innervation is unaffected by the loss of a hair cell, gains a new hair cell and an afferent fiber

that has lost a hair cell and that hair cell is not repaired or replaced, then the value of this replacement pathway is lost.

2- The use of the word plasticity is also somewhat weighted and the authors consider an alternative.

3- The title of the manuscript, focusing on gap junctions as being relevant needs to be changed as this remains the weakest component of the work as the one requiring the most speculation.

Reviewer #2 (Remarks to the Author):

Kindt and colleagues have addressed some of the main concerns raised in the prior review. In particular, they have softened some of their conclusions and addressed whether calcium influx can be induced in hair cells with high K⁺ depolarization, detected with a voltage sensor. As both MET stimulation and high K⁺ exposure are not able to induce presynaptic calcium influx, the findings support the main conclusion that calcium channels are not effectively recruited. The remaining question is why, and this is where the conclusions remain vague. Although there appear to be calcium channels and ribbons in silent cells, as well as postsynaptic specializations, immunocytochemistry is not particularly quantitative, so it isn't clear that the same number of channels are present or that the synapses have achieved the same level of maturation. The lack of recruitment with BayK suggests that far fewer functional channels are present in silent cells. It is also possible that these silent hair cells are either more depolarized, which could cause inactivation of calcium channels, or very hyperpolarized, and therefore would require greater depolarization to trigger activation. As the voltage and K⁺ sensors have not been calibrated, and there are no direct recordings from hair cells, the cause for the lack of recruitment remains speculative, as it isn't clear what the resting potential of the cells is or what level of depolarization is achieved by these stimuli. I remain unconvinced about the direct role of intracellular K⁺ or gap junctions in this phenomenon, as the manipulations are not sufficiently selective. It seems possible that the differences in intracellular K⁺ concentrations among responsive and non-responsive hair cells could be the result of other phenotypic differences between these cells. As noted in the prior review, I think that these experiments add little mechanistic insight to the study and therefore could be removed from the manuscript with no loss in overall impact.

Minor

1. p7 – "To confirm our result" Experiments should not be performed to confirm results.

We appreciate the continued advice and interest of both of the reviewers! In our latest revision, we have made significant modifications to the text as suggested. It is our hope that we have sufficiently addressed your concerns. Main changes include:

- 1. All parts of the manuscript have been modified to de-emphasize and clarify the possible role of gap junctions. In addition, the figures have been rearranged to help place the emphasis on the result that this is a sensory system that functions with many silent connections (pre and post).***
- 2. The discussion has been dramatically modified to incorporate an alternative conclusion (based on an excellent suggestion by the Reviewers) that could account for inactive $Ca_v1.3$ channels.***

The page numbers and line numbers below refer to the unmarked, revised document.

Reviewers' comments:

Reviewer #1 (Remarks to the Author):

I commend the authors for a careful and thorough response to the previous comments on this manuscript. This work represents a significant body of new data that addresses an interesting and novel finding in the neuromast organ suggesting that there are a significant number of mechanically-sensitive hair cells that are synaptically inactive until damage to the neuromast, at which time these cells are recruited into functionality. Data presented that this happens is strong and supports this basic conclusion. However, the arguments about how this happens and what the causal players are in this mechanism remain somewhat confused. The majority of previous comments were addressed directly and clearly and the manuscript is much improved because of it; the remaining issues are as follows.

1- The authors argue that nonactive hair cells may be hyperpolarized so that calcium channels are not active and that K depolarization will not depolarize the cells far enough to activate these channels. Although neither the voltage dye employed nor the K sensor used can quantitatively assess the absolute membrane potentials or K levels, there does seem to be enough data presented to negate this possibility. For example, the 30% difference in K levels measured intracellularly would be predicted to have a 5-10mV effect on resting potential. The high potassium used was perfusing a 1M KCl solution. Aside from the large osmotic effect expected to accompany this perfusion, a change at the hair cell of up to 50 mM would depolarize the hair cell to around -20 mV (ostensibly the maximal calcium current activation potential in hair cells). As access of compounds to the hair cells does not seem to be an issue a 50 mM change representing a 20x dilution of the pipette solution seems pretty conservative. This extremely large change in extracellular K would maximally depolarize the hair cell irregardless of the likely difference in intracellular potassium. Thus hyperpolarization, or input resistance of the cell cannot be responsible for the differences observed. Given that the calcium channels are identified immunocytochemically, the only two real choices appear to be

that the channels are not inserted into the plasma membrane and thus remain insensitive to the electric field changes or that the channels are inserted into the membrane but are biochemically inactivated. Examples of both exist in the literature and can work on the described time frame to switch a cell from synaptically inactive to active.

We agree that the high 1M K⁺ application does seem to be an extreme stimulus. Focal application of K⁺ at concentrations lower than 500uM were unable to penetrate the skin and supporting cells within the time scale of the stimulation (6s). Other drugs do gain access to the hair cells, although this takes approximately 30s when the drug is in the entire bath. We have found that if the skin and excess cells are removed near neuromast organs, lower concentrations of K⁺ (100 mM) can activate hair cells, but with the same overall result.

We also agree that regardless of the exact amount of K⁺ reaching the hair cells, it is likely that hair cells are being depolarized sufficiently to bypass differences in resting membrane potential between active and silent cells - and these depolarizations should be able to activate Ca_v1.3 channels.

Therefore, we have dramatically changed our discussion to entertain the possibility that low [K⁺]_{in} measured in active cells is a consequence of presynaptic function, rather than a prerequisite for function. In this model, Ca_v1.3 channel modifications, interactions or trafficking could render the channels inactive in silent cells (Fig. 8 model 2). Overall, we agree that this is an essential addition to our manuscript.

Pages 18, lines 377-389... Alternatively, it is possible that low hair-cell [K⁺]_{in} levels in active cells is a byproduct of presynaptic function rather than a prerequisite for function (Fig. 8, model 2). This idea is supported by our hair-cell voltage and Ca²⁺ imaging experiments that demonstrate high K⁺ stimulation can depolarize all cells (Fig. 6d-e''') but is unable to activate Ca_v1.3 channels in silent cells (Fig. 6c'''). Because it is difficult to correlate voltage indicator intensity with actual resting potentials and voltage changes, it is not clear from these experiments whether an optimal membrane potential was reached for Ca_v1.3 channel activation in silent cells. If a sufficient depolarization was indeed achieved, then instead it is possible that no amount of depolarization can activate Ca_v1.3 channels in silent cells. If so, then differences in [K⁺]_{in} levels or resting membrane potential are not sufficient to explain inactive Ca_v1.3 channels. Alternatively, Ca_v1.3 channels in silent cells may 1) have different interaction partners, 2) different posttranslational modifications or 3) be improperly localized at the plasma membrane (Fig. 8. model 2). Continued in more detail on lines 389-399.

However, as you and Reviewer #2 pointed out, it is not possible to use this voltage indicator or the K⁺ dye to measure the absolute resting membrane potentials or the absolute changes in voltage. This makes it challenging to completely conclude whether [K⁺]_{in} and resting potential differences play a role in regulating Ca_v1.3 channel activity. In the mammalian cochlea, numerous studies have shown that K⁺ recycling pathways via gap junction systems play an important role in regulating hair-cell activity¹⁻⁴. Therefore, we think that the

observed differences in $[K]_{in}$ between active and inactive cells is an important clue that distinguishes these populations.

Therefore, in our revised manuscript we still postulate that $[K]_{in}$ and resting potentials could impact $Ca_v1.3$ activity (Fig 8. model 1).

Pages 16-17, lines 354-360... In our study, we found that all cells appeared to have properly localized $Ca_v1.3$ channels at presynapses, and only hair-cells with low $[K^+]_{in}$ were associated with presynaptic $Ca_v1.3$ channel activity. Studies on these channels indicate that $Ca_v1.3$ channel open probability and open time occur optimally within a membrane voltage range from -60 to -40 mV^{5,6}. Based on this work, and the Nernst equilibrium potential, increased $[K^+]_{in}$ in silent cells may reflect hyperpolarized membrane potentials that place the cell out of this optimal activation range (Fig. 8, model 1).

2- The gap junctional blockers and Cl channel blocker data remain difficult to reconcile with the fundamental finding of synaptically active and inactive cells. Does the use of these blockers make all cells synaptically active? If there is a causal link between gap-junctional clearance of potassium or Cl channel activity and these two populations of cells than you should see it with the experiments that were performed. It is not clear from the description of the results that this happens.

The block of gap junctions or chloride channels is not satisfying as these drugs actually render nearly all the hair cells inactive rather than activate silent cells (Fig. 5f, Fig S7g'-i). We have tested many compounds and gap junction blockers are the only drugs that raise $[K]_{in}$ levels in hair cells and silence active hair cells. Currently we have not found a class of compounds that can unsilence cells. The link between gap junctions and these two populations is that gap junction block raises $[K]_{in}$ and dramatically reduces synaptic activity even in active cells. By buffering K^+ , gap junctions appear to only clear enough K^+ maintain a subset of hair cell with presynaptic function. We will continue to test this hypothesis via genetic means.

Minor

1- The authors argue that the hair cell recruitment from inactive to active is 'stochastic.' This word has very specific meanings and the data provided does not justify the use of this word.

Stochastic has been removed.

It is unclear how the recruitment can be random, specifically it would be very useful to know if afferent fiber innervation patterns are dictating which cells are activated as activating a hair cell randomly will not necessarily be valuable. If a hair cell whose innervation is unaffected by the loss of a hair cell, gains a new hair cell and an afferent fiber that has lost a hair cell and that hair cell is not repaired or replaced, then the value of this replacement pathway is lost.

Thus far we have not been able to find a pattern that indicates a coherent functional context that is directing which cell is activated after damage. But we agree it would make more sense

if logic based on innervation were dictating the activation. We do know that the afferent neurons are not required to unsilence hair cells (Fig. 5). Whether the activation patterns change without afferents would be interesting to know, but is challenging to determine at this point.

It also is true that randomly unsilencing a hair cell, especially if the new cell is not innervated by the same afferent does not seem like an effective strategy for replacement. Perhaps that is why damage can often unsilence more than one cell (observed in 29 % of trials) – to ensure an appropriate cell is activated. We have added a sentence to the discussion highlighting this:

Pages 19-20, lines 419-422...This suggests that damage acts as a broad cue to unsilence hair cells, without a directive to maintain absolute balance and replacement based strictly on anatomy. Perhaps this is why damage can often (29 % of trials) unsilence multiple cells—to ensure cells of the correct polarity are replaced.

The laser damage we applied experimentally provides very focused damage. It is possible that real world damage to the lateral line is more likely to activate multiple silent cells in order to ensure the correct backup cells are indeed activated.

2- The use of the word plasticity is also somewhat weighted and the authors consider an alternative.

Plasticity has been changed to damage.

3- The title of the manuscript, focusing on gap junctions as being relevant needs to be changed as this remains the weakest component of the work as the one requiring the most speculation.

We completely agree! Please see the new title.

Reviewer #2 (Remarks to the Author):

Kindt and colleagues have addressed some of the main concerns raised in the prior review. In particular, they have softened some of their conclusions and addressed whether calcium influx can be induced in hair cells with high K⁺ depolarization, detected with a voltage sensor. As both MET stimulation and high K⁺ exposure are not able to induce presynaptic calcium influx, the findings support the main conclusion that calcium channels are not effectively recruited. The remaining question is why, and this is where the conclusions remain vague. Although there appear to be calcium channels and ribbons in silent cells, as well as postsynaptic specializations, immunocytochemistry is not particularly quantitative, so it isn't clear that the same number of channels are present or that the synapses have achieved the same level of maturation. The lack of recruitment with BayK suggests that far fewer functional channels are present in silent cells. It is also possible that these silent hair cells are either more depolarized, which could cause inactivation of calcium channels, or very hyperpolarized, and therefore would require greater

depolarization to trigger activation. As the voltage and K⁺ sensors have not been calibrated, and there are no direct recordings from hair cells, the cause for the lack of recruitment remains speculative, as it isn't clear what the resting potential of the cells is or what level of depolarization is achieved by these stimuli.

We agree that it is challenging to make claims regarding resting potentials and voltage changes using dyes and voltage indicators. We now state this in our conclusions before detailing alternatives to $[K^+]_{in}$ influencing synaptic function.

Page 18, lines 381-385...Because it is difficult to correlate voltage indicator intensity with actual resting potentials and voltage changes, it is not clear from these experiments whether an optimal membrane potential was reached for Ca_v1.3 channel activation in silent cells. If a sufficient depolarization was indeed achieved, then instead it is possible that no amount of depolarization can activate Ca_v1.3 channels in silent cells.

We also agree that immunohistochemistry is not always quantifiable, and that it cannot always tell the whole story. If resting potential differences are not influencing Ca_v1.3 channel activity, then it is very possible that Ca_v1.3 channels are rendered inactivate via other, biochemical mechanisms. These alternative mechanisms are outlined below in above in response to Reviewer #1 and below under modification 6.

I remain unconvinced about the direct role of intracellular K⁺ or gap junctions in this phenomenon, as the manipulations are not sufficiently selective. It seems possible that the differences in intracellular K⁺ concentrations among responsive and non-responsive hair cells could be the result of other phenotypic differences between these cells. As noted in the prior review, I think that these experiments add little mechanistic insight to the study and therefore could be removed from the manuscript with no loss in overall impact.

Although we are rigorous in our pharmacology, we agree that these manipulations may not be specific enough. Nevertheless, if innervation is not required for synaptic silencing, we think that the surrounding supporting remain the best candidate to regulate activity and provide a route for communication within the neuromast organ. In addition, including this information could provide insight for other researchers trying to elucidate the role of supporting cells. Therefore, we opt to keep the supporting cell results but modified the manuscript as follows:

- 1. All mention of gap junctions and supporting cells has been removed from the title and abstract.***
- 2. We now more clearly highlight the need for genetics to validate the role of gap junctions.***

Page 17, lines 371-376...Thus far, our present work investigating the role of supporting cells and gap junctions relied solely on pharmacology, and the compounds that block these channels have other targets⁷. Therefore, while the homologues of KIR_{4.1} and Cx26 are expressed in zebrafish

hair-cell organs^{8,9}, additional genetic analyses are needed to determine if they function in zebrafish supporting cells to regulate $[K^+]_{in}$ and hair-cell excitability in neuromasts.

3. **The gap junction and K⁺ data has been combined into a single figure and examples of the gap junctions via TEM are within the supplement.**
4. **The combined gap junction and K⁺ figure (Fig. 5) is now in the center of the manuscript, as just one piece rather than a focus at the end of the paper. Instead the focus is on trying to test if silent cells can be activated – ie high K⁺ (Fig. 6), and ending with the laser damage experiments showing that silent cells can be activated (Fig. 7).**
5. **We state that hair cell $[K_{in}]$ levels could be a byproduct of synaptic function, rather than a cause.**

Page 18, lines 377-381... Alternatively, it is possible that low hair-cell $[K^+]_{in}$ levels in active cells is a byproduct of presynaptic function rather than a prerequisite for function (Fig. 8, model 2). This idea is supported by our hair-cell voltage and Ca^{2+} imaging experiments that demonstrate high K^+ stimulation can depolarize all cells (Fig. 6d-e''') but is unable to activate $Ca_v1.3$ channels in silent cells (Fig. 6c''').

6. **We now outline in detail, an alternative to a role for $[K_{in}]$ levels modulating $Ca_v1.3$ channels – they could be inactivated biochemically.**

Pages 18-19, lines 387-398...Alternatively, $Ca_v1.3$ channels in silent cells may 1) have different interaction partners, 2) different posttranslational modifications or 3) be improperly localized at the plasma membrane (Fig. 8. model 2). Existing literature indicates that all of these scenarios can impact $Ca_v1.3$ channel activity¹⁰⁻¹⁵. For example, the $Ca_v1.3$ pore forming α -subunit has numerous interaction partners including: Ca^{2+} binding proteins (CaBPs), Calmodulin (CaM), and various auxiliary subunits^{10,11,14}. All of these interactors can impact $Ca_v1.3$ channel properties. In addition, phosphorylation of $Ca_v1.3$ channels by PKA or PKG has been shown to promote channel activation¹². Alternatively, $Ca_v1.3$ channels may not be properly trafficked to the plasma membrane. Studies have shown that molecules such as Harmonin and RIM can impact the trafficking of $Ca_v1.3$ channels to the plasma membrane^{13,15}. Additional work is needed to determine if any of these processes could explain why $Ca_v1.3$ channels are not activated in silent cells.

7. **This alternative conclusion is also included in our model(s) (Fig. 8).**

Minor

1. p7 – “To confirm our result” Experiments should not be performed to confirm results. **This has been removed, thank for pointing this out.**

- 1 Cohen-Salmon, M. *et al.* Targeted Ablation of Connexin26 in the Inner Ear Epithelial Gap Junction Network Causes Hearing Impairment and Cell Death. *Current Biology* **12**, 1106-1111, doi:10.1016/s0960-9822(02)00904-1 (2002).
- 2 Martinez, A. D., Acuna, R., Figueroa, V., Maripillan, J. & Nicholson, B. Gap-junction channels dysfunction in deafness and hearing loss. *Antioxid Redox Signal* **11**, 309-322, doi:10.1089/ars.2008.2138 (2009).
- 3 Jagger, D. J. & Forge, A. Connexins and gap junctions in the inner ear--it's not just about K(+) recycling. *Cell Tissue Res* **360**, 633-644, doi:10.1007/s00441-014-2029-z (2015).
- 4 Kikuchi, T., Adams, J. C., Miyabe, Y., So, E. & Kobayashi, T. Potassium ion recycling pathway via gap junction systems in the mammalian cochlea and its interruption in hereditary nonsyndromic deafness. *Med Electron Microsc* **33**, 51-56, doi:10.1007/s007950000009 (2000).
- 5 Koschak, A. *et al.* alpha 1D (Cav1.3) subunits can form l-type Ca²⁺ channels activating at negative voltages. *J Biol Chem* **276**, 22100-22106, doi:10.1074/jbc.M101469200 (2001).
- 6 Xu, W. & Lipscombe, D. Neuronal Ca(V)1.3alpha(1) L-type channels activate at relatively hyperpolarized membrane potentials and are incompletely inhibited by dihydropyridines. *J Neurosci* **21**, 5944-5951 (2001).
- 7 Bodendiek, S. B. & Raman, G. Connexin modulators and their potential targets under the magnifying glass. *Curr Med Chem* **17**, 4191-4230 (2010).
- 8 Chang-Chien, J. *et al.* The connexin 30.3 of zebrafish homologue of human connexin 26 may play similar role in the inner ear. *Hear Res* **313**, 55-66, doi:10.1016/j.heares.2014.04.010 (2014).
- 9 Mahmood, F. *et al.* Generation and validation of a zebrafish model of EAST (epilepsy, ataxia, sensorineural deafness and tubulopathy) syndrome. *Dis Model Mech* **6**, 652-660, doi:10.1242/dmm.009480 (2013).
- 10 Picher, M. M. *et al.* Ca(2+)-binding protein 2 inhibits Ca(2+)-channel inactivation in mouse inner hair cells. *Proc Natl Acad Sci U S A* **114**, E1717-E1726, doi:10.1073/pnas.1617533114 (2017).
- 11 Yang, P. S., Johny, M. B. & Yue, D. T. Allosteric modulation of Ca(2+)-channel by calcium-binding proteins. *Nat Chem Biol* **10**, 231-238, doi:10.1038/nchembio.1436 (2014).
- 12 Mahapatra, S., Marcantoni, A., Zuccotti, A., Carabelli, V. & Carbone, E. Equal sensitivity of Cav1.2 and Cav1.3 channels to the opposing modulations of PKA and PKG in mouse chromaffin cells. *J Physiol* **590**, 5053-5073, doi:10.1113/jphysiol.2012.236729 (2012).
- 13 Picher, M. M. *et al.* Rab Interacting Molecules 2 and 3 Directly Interact with the Pore-Forming CaV1.3 Ca(2+) Channel Subunit and Promote Its Membrane Expression. *Front Cell Neurosci* **11**, 160, doi:10.3389/fncel.2017.00160 (2017).
- 14 Campiglio, M. & Flucher, B. E. The role of auxiliary subunits for the functional diversity of voltage-gated calcium channels. *J Cell Physiol* **230**, 2019-2031, doi:10.1002/jcp.24998 (2015).
- 15 Gregory, F. D. *et al.* Harmonin inhibits presynaptic Ca(v)1.3 Ca²⁺ channels in mouse inner hair cells. *Nat Neurosci* **14**, 1109-1111 (2011).

REVIEWERS' COMMENTS:

Reviewer #1 (Remarks to the Author):

I think the authors have made an admirable attempt to deal with comments from previous versions. The manuscript is much improved from original and previous versions. Although I don't particularly agree with some of the conclusions, I also think the authors have made good efforts and that future work will better elucidate mechanisms going forward. The data presented represent important new findings for the field.